# In Vitro Skeletal Muscle Model of PGM1 Deficiency Reveals Altered Energy Homeostasis

**DOI:** 10.3390/ijms24098247

**Published:** 2023-05-04

**Authors:** Federica Conte, Angel Ashikov, Rachel Mijdam, Eline G. P. van de Ven, Monique van Scherpenzeel, Raisa Veizaj, Seyed P. Mahalleh-Yousefi, Merel A. Post, Karin Huijben, Daan M. Panneman, Richard J. T. Rodenburg, Nicol C. Voermans, Alejandro Garanto, Werner J. H. Koopman, Hans J. C. T. Wessels, Marek J. Noga, Dirk J. Lefeber

**Affiliations:** 1Department of Neurology, Donders Institute for Brain, Cognition and Behavior, Radboud University Medical Center, 6525 GA Nijmegen, The Netherlands; 2GlycoMScan B.V., 5349 AB Oss, The Netherlands; 3Translational Metabolic Laboratory, Department of Laboratory Medicine, Radboud Institute for Molecular Life Sciences (RIMLS), Radboud University Medical Center, 6525 GA Nijmegen, The Netherlands; 4Radboud Center for Mitochondrial Medicine (RCMM), Department of Pediatrics, Amalia Children’s Hospital, Radboud University Medical Center, 6525 GA Nijmegen, The Netherlands; 5Radboud Institute for Molecular Life Sciences (RIMLS), Radboud University Medical Center, 6525 GA Nijmegen, The Netherlands; 6Department of Human Genetics, Radboud University Medical Center, 6525 GA Nijmegen, The Netherlands; 7Department of Pediatrics, Amalia Children’s Hospital, Radboud Institute for Molecular Life Sciences (RIMLS), Radboud University Medical Center, 6525 GA Nijmegen, The Netherlands

**Keywords:** phosphoglucomutase 1, PGM1 deficiency, PGM1 congenital disorder of glycosylation, in vitro muscle model, muscle energy homeostasis, muscle metabolic plasticity

## Abstract

Phosphoglucomutase 1 (PGM1) is a key enzyme for the regulation of energy metabolism from glycogen and glycolysis, as it catalyzes the interconversion of glucose 1-phosphate and glucose 6-phosphate. PGM1 deficiency is an autosomal recessive disorder characterized by a highly heterogenous clinical spectrum, including hypoglycemia, cleft palate, liver dysfunction, growth delay, exercise intolerance, and dilated cardiomyopathy. Abnormal protein glycosylation has been observed in this disease. Oral supplementation with D-galactose efficiently restores protein glycosylation by replenishing the lacking pool of UDP-galactose, and rescues some symptoms, such as hypoglycemia, hepatopathy, and growth delay. However, D-galactose effects on skeletal muscle and heart symptoms remain unclear. In this study, we established an in vitro muscle model for PGM1 deficiency to investigate the role of PGM1 and the effect of D-galactose on nucleotide sugars and energy metabolism. Genome-editing of C2C12 myoblasts via CRISPR/Cas9 resulted in *Pgm1* (mouse homologue of human *PGM1*, according to updated nomenclature) knockout clones, which showed impaired maturation to myotubes. No difference was found for steady-state levels of nucleotide sugars, while dynamic flux analysis based on ^13^C6-galactose suggested a block in the use of galactose for energy production in knockout myoblasts. Subsequent analyses revealed a lower basal respiration and mitochondrial ATP production capacity in the knockout myoblasts and myotubes, which were not restored by D-galactose. In conclusion, an in vitro mouse muscle cell model has been established to study the muscle-specific metabolic mechanisms in PGM1 deficiency, which suggested that galactose was unable to restore the reduced energy production capacity.

## 1. Introduction

Phosphoglucomutase 1 (PGM1) is a mutase responsible for the interconversion of glucose 1-phosphate (Glc-1P) and glucose 6-phosphate (Glc-6P), thereby playing a key role in energy homeostasis by regulating glycogen metabolism and glycolysis (Figure 1). By modulating carbohydrate metabolism, PGM1 is also involved in the synthesis of the precursors of protein glycosylation (Figure 1). In humans, five different α-D-phosphoglucomutase isozymes have been identified (PGM1, -2, -2L1, -3, and -5), although PGM1 is the most broadly expressed and, alone, addresses roughly 90% of the total phosphoglucomutase activity in the majority of human tissues [1,2]. The first genetic variant in *PGM1* gene was reported in 2009 [3] as causative of adult-onset exercise intolerance, which was hypothesized to be linked to disrupted glycogen metabolism, and thus designated as muscle glycogenosis (or glycogen storage disease type XIV) [OMIM:614921]. Nonetheless, abnormal glycogen accumulation in muscle is usually mild and, in some cases, absent [2,3,4,5,6]. In 2012 and 2014 [2,4], it was found that genetic deficiency of PGM1 also caused abnormal protein glycosylation and a much wider spectrum of clinical manifestations including heart, liver, glands, and palate [2,5]. However, the exact molecular mechanisms by which PGM1 regulates processes like glycolysis and protein glycosylation in different affected tissues, such as muscle and heart, remains elusive. 

The identified lack of galactose in protein glycans and the connection of PGM1 to the Leloir pathway (Figure 1) represented the basis of dietary intervention based on galactose in patients [2,6]. On a molecular level, galactose supplementation bypasses the metabolic block posed by PGM1 deficiency by providing an alternative source for Glc-1P synthesis [2,7]. Glc-1P is both the substrate for glycogen synthesis in muscle and hepatic tissues, and the precursor of two building blocks for glycosylation, UDP-glucose (UDP-Glc) and UDP-galactose (UDP-Gal) [2,4,6,7]. Specifically, galactose is metabolized to galactose 1-phosphate (Gal-1P), and then converted to UDP-Gal and UDP-Glc, thus replenishing the depleted pools of these nucleotide sugars even in presence of low Glc-1P levels [2,7].

In clinical practice, galactose supplementation has been administered in patients and proved successful in restoring glycosylation, while rescuing growth delay, reducing hypoglycemic episodes, ameliorating hypogonadotropic hypogonadism, and normalizing liver dysfunction [2,5,7,8]. However, the effect of galactose supplementation on cardiac and skeletal muscle symptoms remained less convincing. Thus far, it has been estimated that over 50% of PGM1-deficient patients present with metabolic myopathy ranging from mild to severe exercise intolerance, muscle weakness, and fatigability, hypotonia, and rhabdomyolysis. In addition, other clinical manifestations reported, although less frequently, include debilitating pain of lower limbs, joints, and tendons (exercise-induced in some cases), recurrent cramps, dysarthria, and myokymia [2,4,5,6,7,8,9,10,11,12,13,14,15,16,17,18]. Hematic tests often revealed elevated serum creatine kinase and myoglobinuria [2,3,5,6,8,9,10,11,12,13,14,15,16,17,18]. Previous studies on the effects of galactose on muscle dysfunction in patients suggested that severe exercise intolerance and weakness partially improved upon galactose exposure [5,6,8], although no amelioration of peak exercise tolerance nor muscle strength occurred.

The current hypothesis based on clinical observations in patients suggests that PGM1 deficiency blocks the final step in the glycogenolytic pathway, resulting in the inability of muscle tissue to use carbohydrates during exercise [3,6]. To compensate for this, the skeletal muscle tissue of patients relies on fatty acids to sustain muscular activity during exercise, as later observed in a few studies [10,19]. Here, development of the so-called ‘second wind’ phenomenon was reported, typical of McArdle disease, including spontaneous fall in exercise heart rate and perceived exertion caused by rapid increases in muscle oxidative capacity sustained not by carbohydrate oxidation (which is blocked) but by exaggerated fat oxidation [19]. Voermans et al. [6] showed that, upon galactose supplementation, the substrate preference of patient’s skeletal muscle is normalized and carbohydrates started being used to sustain exercise. Taken together, these observations suggest that muscle symptoms in PGM1 deficiency may partly respond to galactose supplementation, which may be linked to impaired protein N-glycosylation, but also to reduced energy production capacity in skeletal muscle due to its regulatory role in glycolysis and glycogen metabolism [6,7,10,19].

In this study, we established the first in vitro muscle model for PGM1-linked myopathy via genome editing to investigate the muscle-specific role of PGM1 and the effects of D-galactose on myogenic differentiation, nucleotide sugar metabolism, and energy homeostasis.

## 2. Results

### 2.1. Generation of Pgm1-Knockout C2C12 Myoblasts as Muscle Model for PGM1 Deficiency

To investigate the pathogenic mechanisms underlying myopathy in PGM1 deficiency, we aimed to generate murine *Pgm1* knockout (KO) myoblast lines. Mouse *Pgm1* was targeted with CRISPR/Cas9 for genome editing using different sgRNAs targeting exon 2 or exon 4 (Appendix A), all of which produced viable clones. After selection and expansion, the clones were characterized at molecular level (Sanger sequencing) and tested for PGM enzyme activity to assess successful genome editing. Two *Pgm1*-KO C2C12 myoblast clones showed enzymatic deficiency of PGM with less than 10% activity compared to the wild-type and comparable with levels detected in patient dermal fibroblasts (Table 1) [1]. The effect of *Pgm1* editing was confirmed by Sanger sequencing, which revealed a compound heterozygous ins/del in clone 1, and a homozygous 1 base pair (A) insertion causing frameshift in clone 2 (Appendix A). TA cloning of the compound heterozygous clone revealed a 1 base pair insertion (C) causing frameshift on one allele, and a 15 base pair deletion on the other allele causing an in-frame deletion of five amino acids (Appendix A). We analyzed in silico possible off-targets with up to three mismatches, and of these, we experimentally assessed those containing a PAM sequence (NGG) and located in a gene (either exonic or intronic location). No off-target effect was observed in these clones (Appendix A). After differentiation to myotubes, PGM enzyme activity was further reduced to 5% of the wild-type level (Table 1). Lastly, Pgm1 protein levels were assessed via MS-based proteomics analysis, which confirmed the absence of Pgm1 protein in the KO clones both at myoblast (D-1) and myotube (D7) states (Figure 2b).

### 2.2. Pgm1-Knockout Myoblasts Retain Maturation Potential but Cannot form Elongated Myotubes

As a first step to characterize the effect of PGM1 deficiency on myogenic differentiation, we studied the morphology and differentiation markers upon in vitro myogenic differentiation. At myoblast stage, the morphology of *Pgm1*-KO clones resembled the one of the wild-type, appearing fusiform and elongated with single nucleus and numerous visible nucleoli (Figure 2a and Appendix A). The morphology of mature myotubes at day 7 (D7) of differentiation revealed instead differences. In the wild-type line, the myotubes appeared to be fusing to highly elongated, aligned, and multinucleated fibers with high expression of MyoG (Figure 2a and Appendix A). In *Pgm1*-KO clones the elongated, fibrillar morphology was largely reduced, fusion appeared compromised and MyoG expression was largely decreased (Figure 2a).

Myogenic maturation was also evaluated via gene expression analysis and protein level quantification of muscle maturation markers. Gene expression was assessed via qRT-PCR from samples collected at the myoblast state (D-1, confluence 70%), intermediate state (D3), and myotube state (D7) (Figure 2c). Both KO clones showed increased expression between day 3 to day 7 of both late-differentiation markers *Mef2c* and *Dmd*, indicating that *Pgm1*-deficient myoblasts can mature to some extent, but the formation of elongated and fibrillar myotubes is compromised. This conclusion was further supported by proteomics results, showing that protein levels of a panel of myogenic differentiation markers largely overlapped between wilt-type and KO clones (Figure 2d), although the levels of few main maturation markers, namely Ttn, Myh3, Ryr1, and Casq1, appeared significantly higher in wild-type compared to the two KO clones (Figure 2e).

### 2.3. Steady-State Levels of UDP-Glucose and UDP-Galactose Are Not Depleted in Pgm1-KO Myoblasts and Myotubes

Given the previously reported reduction of UDP-Gal and UDP-Glc in PGM1 deficient fibroblasts and the proposed link to defective glycosylation [2], we first investigated steady-state levels of a range of nucleotide sugars (Figure 3a) in myoblasts and during their differentiation to myotubes via tandem quadrupole ultra-high performance liquid chromatography mass spectrometry (UHPLC-MS). Steady-state analysis of the pool of nucleotide sugars collected during myogenic differentiation (D0, D3, D7), and analyzed via Mann–Whitney test (*p* = 0.05), revealed no significant changes in UDP-Glc levels in *Pgm1*-KO cells as compared to wild-type cells (Figure 3b, Appendix A). Likewise, UDP-Gal levels also did not result significantly reduced in the KO lines in any of the timepoints (Figure 3b, Appendix A). Relative levels of UDP-*N*-acetylhexosamines (UDP-HexNAc), which include both UDP-*N*-acetylglucosamine and UDP-*N*-acetylgalactosamine, consistently increased over differentiation in all three cell lines (Figure 3b), as previously reported in the C2C12 muscle cell model [3]. CMP-*N*-acetyl-neuraminic acid (CMP-Neu5Ac), which can derive from *N*-acetylglucosamine (GlcNAc) 1-phosphate, also resulted increased in the KO lines. In conclusion, no relevant depletion of UDP-Glc and UDP-Gal has been observed in muscle cells, unlike previously reported in primary dermal fibroblasts [2].

### 2.4. Galactose Treatment Does Not Affect Myogenic Differentiation

To evaluate the effects of galactose supplementation on myogenic differentiation, wild-type and *Pgm1*-KO lines were differentiated in the presence of galactose in the culture medium. As the myogenic differentiation medium requires high glucose (Table 2), we designed three treatments with increasing concentrations of galactose (Table 2, Figure 4a), maintaining the final carbohydrate concentration equal to 25 mM to prevent alteration of the differentiation efficiency due to starvation. In the control condition, 25 mM glucose, the wild-type myotubes displayed defined, elongated, and polynucleated fibers consistent with myogenic maturation (Appendix A), confirmed by myosin-4 (Myh4) staining (Figure 4a), while the KO lines showed less defined fibers, with presence of globular polynucleated cells and lower intensity of Myh4 (Figure 2a, Figure 4a and Appendix A). Protein quantification of four myogenic maturation markers further showed support for the more efficient maturation of the wild-type line, since all markers were consistently significantly higher in wild-type than in the KO clones (Figure 4b). The addition of galactose in combination with glucose, both with concentrations of 2 mM and 12.5 mM, did not rescue the maturation efficiency of the KO clones morphologically nor molecularly (Figure 4a,b, Appendix A). On the contrary, galactose addition appeared to reduce Myh4 signal in the KO clones while also reducing the protein levels of Ttn and Ryr1 in KO clone 1 and, less severely, in clone 2. Lastly, all three lines showed signs of failed differentiation when cultured only in presence of galactose (25 mM), in agreement with the literature [4]. Although in all lines some cells survived, no myogenic differentiation and fusion were observed on a morphological level (Appendix A), which was confirmed by the decrease of Myh4 expression assessed via immunostaining (Figure 4a). Moreover, gene expression of Mef2c and Dmd (Appendix A) and protein quantification of Ttn, Myh3, Ryr1, and Casq1, consistently showed a loss of myogenic markers in all three lines differentiated in 25 mM galactose.

### 2.5. Effect of Galactose Treatment on Nucleotide Sugar Levels

The effect of galactose supplementation on nucleotide sugar levels was evaluated by culturing wild-type and *Pgm1*-KO clones for 24 h with differing concentrations of galactose (Figure 5). The first experiment (Figure 5a), conducted in wild-type and *Pgm1*-KO clone 1 myotubes, showed that addition of galactose resulted in a relative increase of UDP-Gal as compared to UDP-Glc in the wild-type line but no clear effect was observed in KO clone 1. In the second experiment (Figure 5b), feeding of wild-type myotubes and *Pgm1*-KO clone 2 myotubes with galactose resulted in a slight increase in the ratio of UDP-Gal/UDP-Glc in both lines. Additionally, the relative levels of UDP-HexNAc decreased upon culturing in 5.5 mM galactose as well as in culture conditions of glucose and galactose, although to a lesser extent. This could be related to the increase of the combined UDP-Glc and UDP-Gal levels or due to a reduced differentiation on galactose. No statistical significance was identified by performing Mann–Whitney test (Appendix A). Taken together, these results show that culturing myotubes in the presence of galactose only resulted in a slight shift in the ratio of UDP-Gal/UDP-Glc.

### 2.6. Dynamic Tracing of ^13^C-Labelled Galactose Shows Reduced Galactose Utilization in Pgm1-KO Clones

To gain more detailed insight into the metabolism of nucleotide sugars in PGM1 deficiency, we performed dynamic analysis of nucleotide sugars using ^13^C-labelled galactose in wild-type and *Pgm1*-KO myotubes. At day 7, the culture medium was replaced with ^13^C6-galactose-containing medium (5 mM Glc + 0.5 mM ^13^C_6_-Gal or 5 mM Glc + 5 mM ^13^C_6_-Gal) and the cell lines were incubated for different time points (5 min, 1, 4, 8, and 24 h) (Table 2, Figure 6).

When cells were cultured in 5 mM of unlabeled Glc combined with low (0.5 mM) or equal (5 mM) concentration of ^13^C_6_-Gal, the label integration into the pools of different nucleotide sugars and four phosphate hexoses behaved similarly (Figure 6). In both cases, the four nucleotide sugars belonging to the biosynthetic branch more proximal to galactose (Figure 3a), namely UDP-Gal, UDP-Glc, UDP-glucuronic acid, and UDP-xylose, reached the isotopic steady state around 4 h in all three lines and in both conditions. However, between 8 and 24 h a metabolic shift occurred in wild-type myotubes but not in the *Pgm1*-KO clones. Specifically, till 8 h the labelled fraction of the pools of these five nucleotide sugars seemed to reach a plateau below 0.3, indicating that less than 30% of the pool of nucleotide sugars are labelled, and thus derived from labelled galactose (while the remaining 70% is unlabeled and likely derived from glucose and from salvage pathways). However, after 8 h the labelled fraction of the pools of these four nucleotide sugars started increasing again reaching 70% labelling at 24 h for UDP-Gal and UDP-Glc. This suggests that wild-type myotubes were able to adapt to the decreasing levels of glucose in the culture medium, by switching to (labelled) galactose as the main source for the de novo synthesis of these four nucleotide sugars. On the contrary, the KO clones did not display the same adaptative capacity (or metabolic plasticity) to switch from glucose to galactose consumption. In fact, both clones seemed to rely on (labelled) galactose to fuel the synthesis of these nucleotide sugars from timepoint 0. Despite the lack of metabolic plasticity in *Pgm1*-KO myotubes, clone 2 displayed a higher labelled fraction in the four sugars at all timepoints, when compared to clone 1. The different dynamics of label integration indicated by the different levels of labelled fraction revealed that clone 2 was more efficient in metabolizing galactose than clone 1. Regarding the other nucleotide sugars that do not belong to the UDP-Glc/UDP-Gal biosynthetic branch, only GDP-Mannose and GDP-Fucose showed some label integration in wild-type myotubes, but almost no label integration in the *Pgm1*-KO clones.

Next, we performed tracer metabolomics to investigate the synthesis of three hexose phosphates, specifically galactose 1-phosphate and glucose 1-/6-phosphate, in the same samples (Figure 6b). In both conditions, the label integration in glucose 1- and 6-phosphate pools in wild-type myotubes mirrors what was observed for the four nucleotide sugars previously discussed: after about 8 h, the wild-type cells are able to switch from glucose to galactose for the synthesis of glucose 1-phosphate (precursor of glycogen synthesis and glycosylation building-blocks) and, to a lesser extent, to glucose-6-phosphate. However, this metabolic plasticity towards glucose 1-phosphate appeared much reduced for both KO clones. In addition, the levels of labelled glucose 6-phosphate in both clones were close to zero, thus confirming the biochemical blocked posed by the loss of function of Pgm1.

Overall, these data indicate that wild-type myotubes possess a metabolic flexibility that allow them to switch between substrates as emergency mechanism to cope with glucose starvation and ensure energy production to meet the cell demand. This metabolic ability is however compromised when PGM1 is deficient, as PGM1 is responsible for the catalysis of the reaction that links galactose and glycogen with glycolysis and energy production.

### 2.7. Pgm1-KO Myoblasts and Myotubes Have Lower Energy Generating Capacity, Not Affected by Galactose Treatment

As a more direct measure to explore the energy production capacity in PGM1 deficiency, we employed the Agilent Seahorse-XFe96 mitochondrial stress test (MST). Prior to this test, we assessed mitochondrial mass and functionality in the different lines across the four culture conditions by quantification of protein levels (Figure 7). Although the total sum of detected mitochondrial proteins suggests an increased mitochondrial content in the two KO myotubes clones (especially in clone 1, Figure 7a), the levels of citrate synthase (Cs) (Figure 7b), cytochrome C (Cyc1) (Figure 7c), and Tomm20 (Figure 7d), considered as markers of cellular oxidative capacity and mitochondrial density [5,6,7,8], appeared similar across all conditions (with the exception of only galactose feeding, which affected cell growth and maturation in all lines but to different extents). Similarly, some of the main subunits of the oxidative phosphorylation complexes [9] showed no significantly different levels across cell lines and conditions, except for only galactose feeding in which cells affected by the non-physiological culture condition (Figure 7c).

Based on the results of the proteomics analysis, we decided to normalize the MST measurements on Cs activity (Figure 8a) and to focus the analysis of KO clone 1, as it represented the clone with the lowest differentiation capacity (Figure 2) and thus the most severe phenotype (Figure 8a). The MST allows to measure key parameters of mitochondrial function by directly measuring the oxygen consumption rate (OCR) of cells, while also monitoring the extracellular acidification rate (ECAR) as an indication of glycolytic activity [10,11].

We analyzed wild-type and *Pgm1*-KO C2C12 cells at myoblast and myotube state under the same four feeding conditions including differing concentrations of galactose (Table 2). Basal respiration, which represents the OCR required to meet the energetic demand of the cell under baseline conditions, was significantly higher in wild-type myoblasts (Figure 8b) and myotubes (Figure 8c) when compared to *Pgm1*-KO clone 1 in all conditions. The presence of Gal, with or without Glc, increased the basal respiration of both wild-type and *Pgm1*-KO myoblasts, whereas in myotubes the basal respiration only increased when Gal was present in combination with glucose. Besides, the OCR remained always higher in the wild-type cells, which suggests that the energy demand of wild-type cells at baseline conditions is higher than the demand in *Pgm1*-deficient cells.

Lastly, we investigated the spare respiratory capacity (SRC), representing the difference between basal respiration (Figure 8b) and the maximal respiration (Appendix A) reached after addition of an uncoupling agent. The SRC characterizes the mitochondrial capacity to meet extra energy requirements (beyond the basal level) in response to acute cellular stress or heavy workload, and thus can be considered a measure of mitochondrial fitness and plasticity [12]. In our data, SRC appeared significantly lower in *Pgm1*-KO myoblasts (Figure 8b) compared to wild-type myoblasts in all conditions, indicating that galactose did not contribute to the normalization of this parameter in *Pgm1*-deficient cells. Conversely, SRC in *Pgm1*-KO myotubes resulted significantly higher than in wild-type myotubes when cells were cultured in only glucose (5.5 mM Glc) (Figure 8c). However, upon supplementation of galactose in culture, the SCR in *Pgm1*-KO myotubes reduced and normalized to the wild-type levels when galactose was present in combination with glucose (5 mM Glc + 0.5 mM Gal; 5 mM Glc + 5 mM Gal). This might indicate that myotubes develop more flexibility in response to critical energy demand, likely due to the fact that myotubes shift towards a more aerobic metabolism and contain a higher number of mitochondria [13,14], as also confirmed by our proteomics analysis (Figure 7a).

OCR levels linked to proton leak and non-mitochondrial respiration were reduced or similar between wild-type and *Pgm1*-KO myoblasts (Appendix A) and did not show clear effects from galactose supplementation in the culture conditions.

Related to glycolytic activity, ECAR measurement suggested that glycolysis was less active in *Pgm1*-KO myoblasts compared to wild-type myoblasts in all conditions, except in the condition based on galactose alone (5.5 mM Gal) in which the two cell lines displayed overlapping ECAR (Figure 8d). ECAR differences between *Pgm1*-KO myotubes and wild-type myotubes were more subtle in all conditions, with the exception the only galactose feeding (5.5 mM Gal) in which KO myotubes displayed a lower ECAR relative to wild-type myotubes (Figure 8d). In the other conditions, the ECAR at baseline was similar in both lines, but upon addition of oligomycin (which blocks the mitochondrial FoF1-ATPase), the KO myotubes showed increased acidification that suggests a shift towards anaerobic energy production. After the first addition of the uncoupling agent, the ECAR levels of KO myotubes again become similar or lower than the levels in wild-type myotubes. These data hint towards the possibility that, at myotube state, *Pgm1*-KO cells better respond to altered mitochondrial energy production, likely also due to the glucose still present in the XF minimal DMEM that is added to the cells 1 h before the OCR/ECAR measurements. We hypothesize that more substantial differences could emerge in KO myotubes in case of complete glucose starvation, when the cells are pushed towards using galactose and glycogen (both flowing via PGM1) to meet their energy demand. However, due to the set-up of this analysis, we could not avoid using the Seahorse medium for analysis which contains 5.5 mM glucose.

Overall, these data suggest that *Pgm1*-KO clone 1 displays a reduced metabolic plasticity and a lower efficiency in mitochondrial ATP production, supporting the presence of an energetic deficit linked to PGM1 deficiency.

## 3. Discussion

The pathogenic mechanisms underlying the development of myopathy and associated muscle defects in PGM1 deficiency remain elusive, and may be related to abnormal protein glycosylation as well as disturbed energy production. Most of the biochemical and molecular studies on the pathogenesis of PGM1 deficiency have thus far been conducted in primary dermal fibroblasts derived from patients which do not properly reflect the pathogenesis of tissue-specific mechanisms in muscle. In this study, we established the first in vitro muscle-specific model for PGM1 deficiency by knocking out mouse *Pgm1* (homologue of human *PGM1*) in C2C12 myoblasts (Figure 2). Unlike previously reported in dermal fibroblasts [2], our data in myoblasts and myotubes show no depletion of UDP-Glc and UDP-Gal in both muscle cell types, likely due to the effect of glycogen on the synthesis of these two nucleotide sugars (Figure 3). Moreover, no major effects of galactose supplementation on myogenic differentiation or on the levels of nucleotide sugars were observed (Figure 4 and Figure 5). However, dynamic tracing of ^13^C_6_-galactose in myotubes suggested a defect in galactose utilization for energy production (Figure 6). This was confirmed by the Seahorse-based MST which indicated reduced ATP production and basal respiration in both myoblasts and myotubes, not rescued to physiological level by galactose supplementation (Figure 8). Both *Pgm1*-KO clones have similar residual enzyme activity and behave similarly in dynamic tracing, but appear to have differences in the efficiency of uptake and utilization of galactose as highlighted by the isotopic labelling experiment (Figure 6). This clonal diversity might be due to variation in the molecular background (e.g., modifiers that have been favored due to single cell clone expansion), differences at the epigenetic level or caused by unpredicted collateral editing of off-targets by CRISPR/Cas9 methodology. However, predicted off-targets were tested and no alterations were found (Appendix A).

Based on the current data, Pgm1 deficiency in skeletal muscle cells poses a block in the UDP-Gal/UDP-Glc biosynthetic branch bridging galactose and glycogen with glycolysis, which has been confirmed by tracer metabolomics performed on Glc-1P and Glc-6P (Figure 6b).

Aside from UDP-Gal/UDP-Glc synthesis, galactose can flow into glycolysis when Pgm1 is available, but does not effectively contribute to ATP production. In fact, the net gain of ATP from glycolysis is two molecules of ATP. However, galactose requires two molecules of ATP to be converted to glucose 6-phosphate and enter glycolysis, thus its consumption does not result in a net gain of ATP, forcing the cells to activate aerobic energy production [15,16,17]. For this reason, in vitro galactose feeding is used to counterbalance the Crabtree effect and consequent suppression of oxidative phosphorylation in human muscle cells [15,16,18]. However, Elkalaf et al. [4] discovered that wild-type C2C12 muscle cells do not respond in such a way to full galactose replacement in culture conditions, but rather respond to low glucose feeding. This study aligns with our results from the Seahorse-MST, in which mitochondrial ATP production and other bioenergetic parameters indicative of the mitochondrial oxidative metabolism in C2C12 myoblasts (both wild-type and *Pgm1*-KO) appears lower when the cells are fed with low glucose (5.5 mM) than when fed with only galactose (Figure 8b,c). This could explain why in our model galactose supplementation only slightly improves basal respiration and mitochondrial ATP production in the KO clone, but fails in reaching wild-type levels, in both myoblasts and myotubes (Figure 8b).

Besides flowing into glycolysis, in muscle cells galactose can fuel into the galactonate pathway, thus contributing to the production of cytosolic NAD production that can be converted to ATP by the oxidative phosphorylation, or can be used to modulate chromatin modifiers, such as SIRT1 deacetylase, to translate the metabolic changes into epigenetic and genetic regulation [19,21]. Although galactonate is mostly detected in pathological conditions, such as in patients affected by galactosemia [OMIM#230400], further investigations should be planned in this direction to evaluate its presence and contribution in PGM1 deficiency.

Based on our data, we speculate that aberrant energy homeostasis in PGM1-deficient skeletal muscle might translate in a less efficient contraction machinery, as ATP is necessary to trigger myosin-actin binding [15,16]. If confirmed in future studies in human muscle models, this hypothesis would explain the development of myopathy, exercise intolerance, muscle weakness, and other manifestations in PGM1-deficient patients. Besides, our preliminary findings asuggest that defective muscle energy homeostasis is further enhanced in presence of low glucose in culture, as PGM1-deficient myotubes appear unable to rewire efficiently their metabolism in order to fuel alternative carbohydrate substrate (such as galactose) into ATP production (Figure 6). This lack of metabolic plasticity that manifests in low glucose conditions could explain why in some patients frequent-feeding regimen and rescuing of hypoglycemic episodes (via galactose supplementation) helped reducing the frequency and severity of muscle weakness and other muscular complaints, not as a direct effect of galactose metabolization but as a secondary effect of stabilization of hematic glucose levels [22,23,24].

In conclusion, we established the first in vitro muscle model that enabled us to perform muscle-specific investigation of PGM1-linked myopathy. Furthermore, we provided preliminary evidence of a lower metabolic plasticity and defective energy homeostasis as additional pathogenic mechanisms underlying muscle dysfunction in PGM1-deficient patient. In absence of lipids in the culture medium, we have shown that galactose is not able to normalize the energy homeostasis nor to restore the metabolic flexibility of PGM1-deficient C2C12 cells. If our findings will be corroborated by future studies in human muscle models, in the future other metabolites beyond galactose should be investigated as potential targets for treatment design to address metabolic myopathy in PGM1-CDG patients.

## 4. Materials and Methods

### 4.1. Genome Editing of C2C12 Myoblasts via CRISPR-Cas9

Murine immortalized C2C12 myoblasts (American Type Culture Collection, Rockville, MD, USA) were used to generate *Pgm1*-deficient muscle cell models. Six synthetic oligos were designed via CRISPR Design Tool (http://tools.genome-engineering.org, accessed on 5 April 2018) as well as CHOPCHOP (https://chopchop.cbu.uib.no/, accessed on 5 April 2018) and CRISPOR (http://crispor.tefor.net/, accessed on 5 April 2018) [25] and used to generate 3 sgRNA targeting exons 2 and 4 of the *Pgm1* mouse gene (Appendix A). Due to recent changes in the official genetic nomenclature, the mouse homolog of human *PGM1* that used to be called *Pgm2* has now been renamed as Pgm1 (ENSMUSG00000025791). The sgRNAs were hybridized and cloned into the PSpCas9(bb)-2A-GFP (PX458) vector containing GFP and Cas9 coding sequences (donated by Dr. F. Zhang). C2C12 myoblasts were transfected using lipofectamine 2000 (ThermoFisher Scientific, Waltham, MA, USA) according to manufacturer instructions. Next, cells were grown for 48 h after which GFP-positive cells were selected by flow cytometry. GFP-positive cells were seeded in a 96-well plate 1 cell/well. The resulting single colonies were collected and further cultured to establish clonal lines. Subsequently, Sanger sequencing was conducted to determine the successfully edited clones (Appendix A). Successful homozygous or compound heterozygous clones were used for subsequent PGM enzyme activity assay (Section 4.4). To identify the allelic mutations of compound heterozygous clones, TA cloning was performed using pGEM-T Easy Vector System (Promega, Madison, WI) according to manufacturer’ instructions. In addition, analysis of the off-targets most likely to occur was performed. We prioritized based on the likelihood of their occurrence dependent on (i) the number of mismatches (0 to 3), (ii) the presence of the PAM sequence, and (iii) the genic vs. intergenic location (Appendix A). The primers for assessment of the selected off-targets were designed via Primer-BLAST (https://www.ncbi.nlm.nih.gov/tools/primer-blast/, accessed on 25 January 2019) and used for PCR amplification using standard conditions and Sanger sequencing (Appendix A).

### 4.2. C2C12 Myoblast Culture and Differentiation to Myotubes

Murine C2C12 myoblasts, both wild-type and *Pgm1*-KO lines, were cultured in high glucose DMEM (Gibco, Life Technologies, Carlsbad, CA, USA), including 25 mM glucose, 1 mM sodium pyruvate and 3.97 mM L-glutamine, supplemented with 10% of fetal bovine serum (FBS) (Gibco, Life Technologies, Carlsbad, CA, USA) and 1% Penicillin/Streptomycin (Gibco, Life Technologies, Carlsbad, CA, USA) (Table 2). Myoblasts were cultured in T80 flaks and passaged when reaching 60–70% confluence (about every 48 h) to prevent spontaneous differentiation.

Prior to differentiation, myoblasts were seeded in 6-well plates at 5.5 × 10^4^ cells/well. Myogenic differentiation towards myotubes was induced by replacing the culture medium with differentiation medium after allowing myoblasts to reach full confluence. The differentiation medium was based on high-glucose DMEM (Gibco, Life Technologies, Carlsbad, CA, USA), supplemented with 2% FBS (Gibco, Life Technologies, Carlsbad, CA, USA) and 1% Penicillin/Streptomycin (Gibco, Life Technologies, Carlsbad, CA, USA) (Table 2). During differentiation, the medium was refreshed every 24 h for a period of 7 days. The differentiation was monitored daily via optic microscopy (optic microscope Zeiss Axiovert25 equipped with Canon camera and images were processed using FIJI/ImageJ2 v.1.8.0_322).

From day 7 onwards, myotubes were cultured in low glucose DMEM (Gibco, Life Technologies, Carlsbad, CA, USA), including 5.5 mM glucose, 1 mM sodium pyruvate, 3.97 mM L-glutamine, supplemented with 2% FBS (Gibco, Life Technologies, Carlsbad, CA, USA) and 1% Penicillin/Streptomycin (Gibco, Life Technologies, Carlsbad, CA, USA).

The myoblast and myotube culture media used for culturing cells meant for mitochondrial stress assay (Section 4.8) application were not supplemented with 1 mM sodium pyruvate (Table 2).

### 4.3. Immunofluorescence Staining

Myotubes were cultured in 12-well plates on borosilicate cover-glasses for 7 days with the different types of differentiation media (Table 2). Once ready, the cells were washed twice with PBS 1X (Gibco, Life Technologies, Carlsbad, CA, USA), and then incubated for 15 min at room temperature with 1 mL/well of 4% PFA as fixative solution (Sigma-Aldrich, Saint Louis, MO, USA). The fixed cells were washed three times with 1 mL/well of PBS 1X (5 min/wash), and the cover-glasses were then preserved in 700 μL per well of glycerol 50% (Sigma-Aldrich, Saint Louis, MO, USA) at 4 °C until staining. For staining, the glycerol solution was removed by six washing steps each using 1 mL/well of PBS 1X (1 min/wash). Next, the cells were permeabilized by addition of 1 mL/well of 0.1% Triton X-100 in 3% BSA/1X PBS solution, and incubated for 10 min at 4 °C. The coverslips were then washed three time with PBS 1X (5 min/wash), as done before permeabilization, and then incubated with 1 mL/well of 3% BSA/1X PBS (blocking solution) for 30 min at room temperature. After three more washes in PBS 1X, cells were incubated with the primary antibodies against Myosin-4 (eBioscience, ThermoFisher Scientific, Waltham, MA, USA), diluted 1:100, for 1 h at room temperature in wet chamber. Next, the cover-glasses were washed three times in 100 μL 0.05% Tween-20 in 1X PBS (10 min/wash), and then incubated with the secondary fluorochrome-conjugated antibodies Alexa 647 goat anti-mouse IgG2b (Invitrogen, Thermo Fisher Scientific, Waltham, MA, USA) diluted 1:350, for 1 h at room temperature in the dark. The over-glasses were then washed again three times in 100 μL 0.05% Tween-20 in 1X PBS, and then incubated with 100 μL of 0.75% DAPI solution in PBS 1X (Molecular Probes, Life Technologies, Carlsbad, CA, USA) for 6–7 min in the dark. Lastly, the cover-glasses were washed with milliQ water and mounted on microscope slides using ProLong™ Diamond Antifade Mountant (Molecular Probes, Thermo Fisher Scientific, Waltham, MA, USA). After 24 h, pictures were taken using a Zeiss Axio Imager Z2 Upright Microscope, and processed using FIJI/ImageJ2 v.1.8.0_322.

### 4.4. Phosphoglucomutase Enzyme Activity Assay

Phosphoglucomutase activity was measured in cell lysates from myoblasts (collected at day 0) and from myotubes (collected at day 7) on a Konelab 20XTi (Thermo Scientific Scientific, Waltham, MA, USA). Cell pellets were prepared by removing the medium from the wells, washing the cells with 1X PBS (Gibco, Life Technologies, Carlsbad, CA, USA), detaching them with Trypsin/EDTA solution, and collecting them in 15 mL Falcon tubes. Next, the cells were pelleted down via centrifugation at 550× *g* for 5 min at room temperature. The pellets were then washed with 6 mL of 0.9% NaCl three times. After the centrifugation for the last washing step, the supernatant was aspirated from the Falcon tubes, and the dry pellets were snap-frozen in liquid nitrogen and stored at −80 °C. The enzyme activity assay is based on the quantification of NADPH production, measured by absorbance at 340 nm according to Van Schaftingen and Jaeken [20]. Data on PGM activity were normalized on protein content, measured also via Konelab 20XTi machine.

### 4.5. Gene Expression of Myogenic Maturation Markers

#### 4.5.1. RNA Extraction and cDNA Synthesis

RNA was extracted from cells collected at different timepoints during myogenic differentiation: day −1 (myoblast state), day 3 (intermediate state), day 7 (myotube state). Myoblasts were cultured in 145 mm diameter Petri dishes, to allow to reach a sufficient number of cells for RNA extraction while avoiding high confluence, which could trigger spontaneous differentiation. Cells collected at other timepoints were cultured in 6-well plates, as they are expected to reach full confluence. Prior to extraction, cells were lysed in 1 mL of TRIzol reagent (Ambion, Thermo Fisher Scientific, Waltham, MA, USA), snap-frozen in liquid nitrogen and stored at −80 °C until extraction. RNA extraction was performed with Direct-zol RNA MiniPrep Plus kit (Zymo Research, Irvine, CA, USA), according to manufacturer instructions. After extraction, 1 µg of total RNA was used for cDNA synthesis using RevertAid First Strand cDNA Synthesis Kit (Thermo Fisher Scientific, Waltham, MA, USA), according to manufacturer instructions.

#### 4.5.2. RT-qPCR of Myogenic Maturation Markers

cDNA samples were diluted 10X with nuclease-free water (Promega, Madison, WI) and used for quantitative real-time PCR (qRT-PCR) expression analysis of *Mef2c* (late differentiation) and *Dmd* (myotube marker). The *Ap3d1* gene was chosen as house-keeping gene for comparison of gene expression and normalization. The details of the primers used are reported in Appendix A. qRT-PCR was performed in triplicate using the GoTaq PCR Master Mix (Promega, Madison, WI) according to manufacturer instructions. The 96-well PCR plates were then loaded in the BioRad CFX96 Real-Time PCR thermocycler and detection system (BioRad, Hercules, CA, USA). The amplification was measured based on SYBR Green (Promega, Madison, WI, USA). The C_t_ (also referred to as C_q_) values were recorded by BioRad CFX manager software (BioRad, Hercules, CA, USA). Data analysis was based on relative gene expression analysis using the 2^−ΔΔCt^ method by Livak & Schmittgen [26]. In this comparative method, the expression levels of target genes are normalized based on the house-keeping gene *Ap3d1*. Statistical analysis and data visualization were performed using PRISM GraphPad v9.4.1.

### 4.6. Mass Spectrometry-Based Proteomics

#### 4.6.1. In-Solution Digestion and Sample Loading

Myoblasts and myotubes were collected from 6-well plates (1 well = 1 extract) via EDTA incubation and pelleted in PSB 1X. Dry pellets were diluted in 40 μL (1:1) of 8 M Urea/10 mM Tris pH 8.0, followed by addition of 40 μL reduction buffer (10 mM dithiothreitol in mQ) and incubation for 30 min at room temperature. Subsequently, the samples were alkylated with 40 μL of 50 mM 2-chloroacetamide (CAA) in 50 mM ammonium bicarbonate (ABC) and incubated for 20 min at room temperature in the dark. After, we added 80 μL of ABC to each sample, and digested them by the addition of Lys-C 5 µL/50 µg protein (Promega, Madison, WI, USA) and trypsin 5 µL/20 µg protein (Promega, Madison, WI, USA) endopeptidases overnight at 37 °C. Next, the digested samples were diluted 4 times with 50 mM ABC and further digested with the addition of trypsin (1 µL/50 µg protein) and incubated overnight at 37 °C. The next day, samples were filtered with Amicon^TM^ ultra-0.5 30 KDa centrifugal filters (Merck, Darmstadt, DE, USA), and protein concentration was determined for each clarified extract upon loading in EVOtips (Evosep, Odense, DK, USA), to ensure the loading of a protein amount of 100 ng (in 20 uL) for each tip. This extraction and digestion method is mostly effective for soluble proteins or membrane proteins with large extramembrane domains.

#### 4.6.2. Liquid Chromatography-Mass Spectrometry/Mass Spectrometry (LC-MS/MS) Analysis

The digested samples were injected in a timsTOF Pro 2 (Bruker Daltonics, Bremen, DE, USA), coupled to an Evosep One LC system. The Evosep system was connected to a C18 analytical column (nanoElute FIFTEEN, PepSep 8 cm, 100 µm I.D., 1.9 µm particle size) using the pre-programmed gradient for 100 samples per day. TimsTOF Pro 2 was operated in positive Parallel Accumulation-Serial Fragmentation (PaSEF) mode (Bruker Daltonics, Bremen, DE, USA). The MS and MS/MS spectra were recorded from m/z 300–1800 with a mobility range of 0.6–1.6 K0 (1/K0), and the accumulation and ramp time of 100 ms. Data-dependent acquisition (DDA) was performed using 10 PaSEF MS/MS scans per cycle with 100% duty cycle, pre-pulse storage time of 12.0 µs, and transfer time of 60 µs. Active exclusion time of 0.4 min was applied as well as precursors that reached 20,000 intensity units. Collision cell RF was set to 1500 Vpp and collision energy was ramped as a function of the ion mobility.

LC-MS/MS datasets were real-time analyzed with parallel database search engine in real-time (PaSER). For all searches a reviewed Mus musculus protein database including contaminants was used (Uniprot, 17137 entries, downloaded on 16 December 2022). A decoy database was generated and added to this database. Precursor/peptide mass tolerance was set to 20 ppm and fragment mass tolerance was set to 30 ppm. Precursor mass range was set between 600.0 and 6000.0 Da with a charge range of 0–1000. Tryptic cleavage specificity was set with a maximum of 2 missed cleavages. Minimum peptide length was set to 6 with a maximum of 500 residues. Static modification was set for carbamidomethyl cysteine modifications. Variable modifications were set for protein N-terminal acetylation and methionine oxidation. Maximum number of scans in split spectral files was set to 4000. ProLuCID primary score type was set to XCorr with secondary type Z-score. MS/MS spectra were not deisotoped and decharged and low fragment ions were not used. The datasets were label-free quantified without match between runs (MBR). Mass tolerance was set to 10 ppm. The retention time at 30 sec and the ion mobility tolerance at 0.03 1/K0. The maximum RMSE Isotopid Ratio was set at 0.2 identified peptides and missing values. After PaSER processing, data were exported and analyzed in PRISM GraphPad v9.4.1. Based on normality test results, significance was tested via one-way ANOVA with Bonferroni post hod correction for multiple comparisons.

### 4.7. Mass Spectrometry-Based Analysis of Nucleotide Sugars Levels and Metabolic Flux

#### 4.7.1. Polar Metabolite Extraction

To extract the polar metabolites after incubation of the myoblasts or myotubes with the treatment media according to the specifics of each experimental design (Table 2), the 6-well plates were washed twice with 2 mL/well of 75 mM ammonium carbonate solution (Honeywell, Charlotte, NC, USA), with pH adjusted to 7.3 (±0.1) with acetic acid. After removing the washing solution, the plates were snap-frozen by contact of the underside of the plate with liquid nitrogen and stored at -80 °C until extraction.

Polar metabolites were extracted using a −20 °C cold extraction buffer of 40:40:20 *v*/*v* Acetonitrile (Biosolve, Valkenswaard, NL, USA), Methanol (Honeywell, Charlotte, NC, USA) and ultra-pure Versol water (Laboratoire Aguetant, Saint-Fons, France). While keeping the plates on ice, each well was incubated for 5 min with 1.4 mL of extraction buffer at 4 °C. Next, the extracts were centrifuged at 13,000 rpm for 3 min at 4 °C to pellet and remove cell debris. Lastly, the samples were dried in a vacuum centrifuge (Savant SC100, RVT 100 with an external oil pump) overnight at room temperature and stored at −80 °C until analysis.

#### 4.7.2. Analysis of Nucleotide Sugars by Ion-Pair LC-MS and Data Processing

The metabolite extracts were dissolved in 100 μL of milliQ water and 10 μL were loaded into an Agilent 1290 liquid chromatography mass spectrometer (LC-MS). Steady-state (unlabeled) nucleotide sugar profiling was performed as previously described [3]. Tracing of ^13^C_6_-galactose into nucleotide sugars and hexose phosphates was performed using, respectively, the TEA-based and the TBA-based LC-MS methods as reported in van Scherpenzeel et al. [3], but adapted to include transitions for the labeled analogs of each nucleotide sugar. For the tracer metabolomics in nucleotide sugars the isotopologues M+6 and M+11 were measured to monitor the direct flux of ^13^C_6_-Gal into the hexose part of the respective UDP-sugars (Appendix A). For the hexose phosphates, the M+6 isotopologue was measured (Appendix A). Agilent 6490A QQQ mass spectrometer was operated in dynamic multiple-reaction monitoring (MRM) mode. The lists of transitions used to detect isotope incorporation in the targeted compounds are reported in Appendix A. LC-MS data were processed in Skyline (20.2) using a target list chosen on the basis of the *in silico*-generated transition list (Appendix A). Resulting peak areas were imported in Microsoft Excel and normalized (i) to the sum of all nucleotide sugars detected to obtain relative abundances in steady-state experiments, or (ii) to the ratio of labelled vs. total fractions in the metabolic flux studies (^13^C/^12^C + ^13^C) [27]. Data processing and visualization were performed in PRISM GraphPad v9.4.1, and the significance of unlabeled metabolomics data was tested via Mann-Whitney test (*p* = 0.05).

### 4.8. Analysis of Mitochondrial Respiration and Bioenergetics Parameters

#### 4.8.1. Mitochondrial Stress Test

In order to apply the Mitochondrial Stress Test (MST) in the Seahorse XFe96 Analyzer (Agilent Technologies, Santa Clara, CA, USA), titration of the carbonyl cyanide-p-trifluoromethox-yphenyl-hydrazon (FCCP, Sigma-Aldrich, Saint Louis, MO, USA) was necessary to establish the concentration allowing to reach the highest oxygen consumption rate (OCR) [10,11]. The titration was performed at myoblasts and myotube states for both lines (wild-type line and compound heterozygous KO-line clone 1), and for each treatment media (Appendix A). Along with the OCR, the extracellular acidification rate (ECAR) was also monitored and recorded as an indicator of glycolytic activity.

For the experiments, myoblasts were seeded in the Seahorse XFe96 cell culture plate (5 × 10^3^ cells/well) and incubated at 37 °C and 5% CO_2_. Myoblast culture medium was replaced with the treatment media (Table 2) after 24 h and incubated with such media for 24 h at 37 °C and 5% CO_2_. Myotubes were generated by differentiation for 7 days in the Seahorse XFe96 culture plate by refreshing them with differentiation medium every 48 h. On day 7, the differentiation medium was replaced with the treatment media (Table 2). After 24 h of incubation in the treatment media, the cells were moved to the Seahorse assay medium, named XF base medium (Agilent minimal DMEM including L-glutamine, pyruvate, and glucose) upon adjusting its pH to 7.4 using 1 N NaOH. Then, the culture plate was incubated at 37 °C without CO_2_ for 1 h.

The Seahorse sensor cartridge (previously hydrated overnight) was loaded with 20 μL of 1.0 μM oligomycin in port A, 22 μL of 0.5–3 μM FCCP in port B (Appendix A), 25 μL of 0.5/1.5 μM FCCP in port C and 27 μL of 50 μM rotenone/antimycin-A solution [10,11]. Oligomycin is an inhibitor of complex V (ATP synthase) of the mitochondrial respiratory chain. FCCP acts as an ionophore that dissipates the gradient uncoupling electron transport from complex V activity and increasing oxygen consumption to a maximum level. Lastly, rotenone and antymicin-A are inhibitors of complexes I and III, respectively, that are able to completely halt the respiratory chain [11,28]. By sequential injection of these inhibitors, Seahorse XFe96 analyzer is able to measure the alteration of the OCR from which different parameters can be derived: basal respiration, mitochondrial ATP production, proton leak, non-mitochondrial respiration, maximal respiration, and spare respiratory capacity [11].

After the OCR (and ECAR) measurement, the plate was retrieved, the medium was removed and 20 μL of 0.33% Triton/TrisHCl with pH of 7.4 was added to each well. The plate was then stored at −80 °C to assess citrate synthase activity. Data acquisition was performed via Agilent Wave Controller software 2.6.

#### 4.8.2. Citrate Synthase Activity Assay for Data Normalization

The MST results were normalized using the citrate synthase (Cs) activity assay performed on the same plates. After the MST was performed on the Seahorse XFe96 Analyzer, the plate stored at −80 °C underwent multiple cycle of freeze–thawing in liquid nitrogen to lyse the cells. Next, 20 μL of 3 mM Acetyl Coenzyme-A solution (Acetyl-CoA lithium salt, Sigma-Aldrich, Saint Louis, MO, USA) and 150 μL of CS-activity mix [28] were added to each well. After performing a blank measurement, 10 μL of 10 mM oxaloacetic acid (Sigma-Aldrich, Saint Louis, MO, USA) in 0.1 mM Tris was added to each well and the absorbance recorded on a Spark Multimode Microplate Reader (Tecan, Männedorf, CH, USA). Statistical significance was evaluated via one-way ANOVA with Bonferroni post hoc correction for multiple comparisons (*p* = 0.05). The normalization of MST results was performed in Microsoft Excel, while data processing and statistical analysis were performed using PRISM GraphPad v9.4.1.

## Figures and Tables

**Figure 1 ijms-24-08247-f001:**
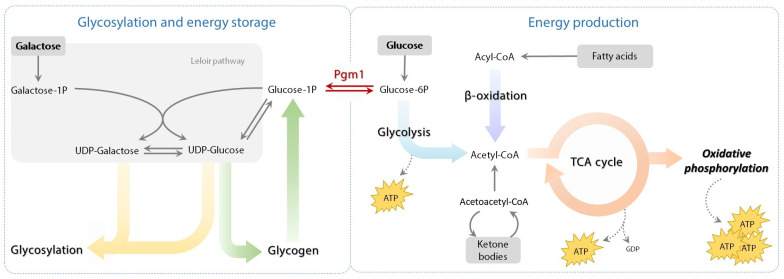
Biochemical network of PGM1 in muscle. Schematic representation of the PGM1-mediated reaction in between protein glycosylation and energy production in the context of skeletal muscle.

**Figure 2 ijms-24-08247-f002:**
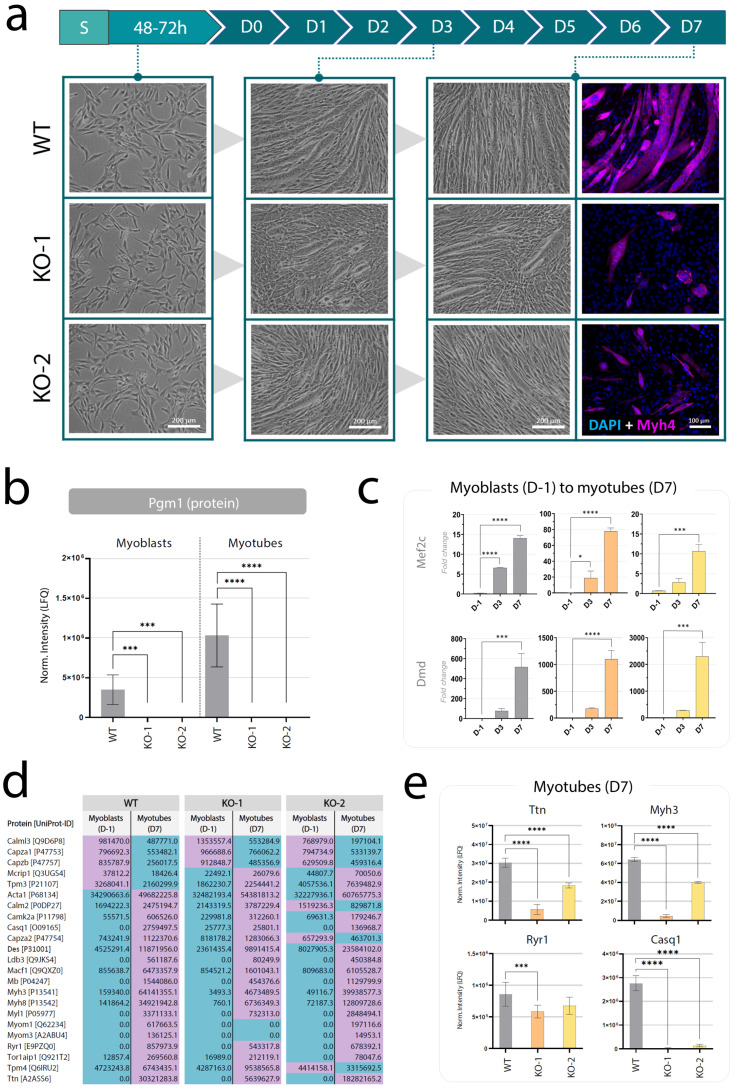
Characterization of Pgm1-KO C2C12 cells. (**a**) Timeline of the myogenic differentiation. The myotubes are seeded (S) and, after reaching full confluence (48–72 h), differentiation is initiated (D0) and completed after 7 days (D7). Pictures in bright-field of the C2C12 lines have been collected 24 h before starting differentiation (D-1, myoblast state, <70% confluent), at the intermediate state (D3) and after differentiation (D7, myotube state) for the wild-type line and for the two *Pgm1*-KO clones. Slides for immunofluorescent staining were collected at D7 (myotube state) and stained with DAPI (blue, nuclear) and anti-Myosin 4 (magenta, cytoplasmic). Scale bars: 200 µm (brightfield), 100 µm (staining). More pictures are provided in Appendix A. (**b**) Pgm1 protein levels in wild-type and KO myoblasts (D3) and myotubes (D7), normalized via label-free quantification (LFQ) and processed via one-way ANOVA with Bonferroni post-hoc correction for multiple comparisons (only significant differences are shown: *** *p* < 0.001, **** *p* < 0.0001; n = 10). (**c**) Expression of muscle maturation markers monitored before (D-1), during (D3), and after differentiation (D7) in wild-type cells and *Pgm1*-KO clones. The relative expression (expressed as fold change) is normalized to the expression of the house-keeping gene *Ap3d1* at timepoint 0. Error bars represent the standard deviation of 2^–ΔΔCq^ (n = 3) and significance has been determined via one-way ANOVA with Bonferroni post-hoc correction for multiple comparisons (only significant differences are shown: * *p* < 0.05, *** *p* < 0.001, **** *p* < 0.0001; n = 3). (**d**) Protein levels of a panel of myogenic differentiation markers measured in each line at D-1 (myoblast state, <70% confluent) and D7 (myotube state) normalized via LFQ Blue: lower level; purple: higher level (n = 10). The results of MS-based proteomics analysis are reported in Appendix A. Legend: blue, lower level; purple, higher level. (**e**) Comparison of normalized protein levels of main myogenic markers across wild-type and KO myotubes (D7). Significance was tested via one-way ANOVA with Bonferroni post hod correction for multiple comparisons (only significant differences are shown: *** *p* < 0.001, **** *p* < 0.0001; n = 10).

**Figure 3 ijms-24-08247-f003:**
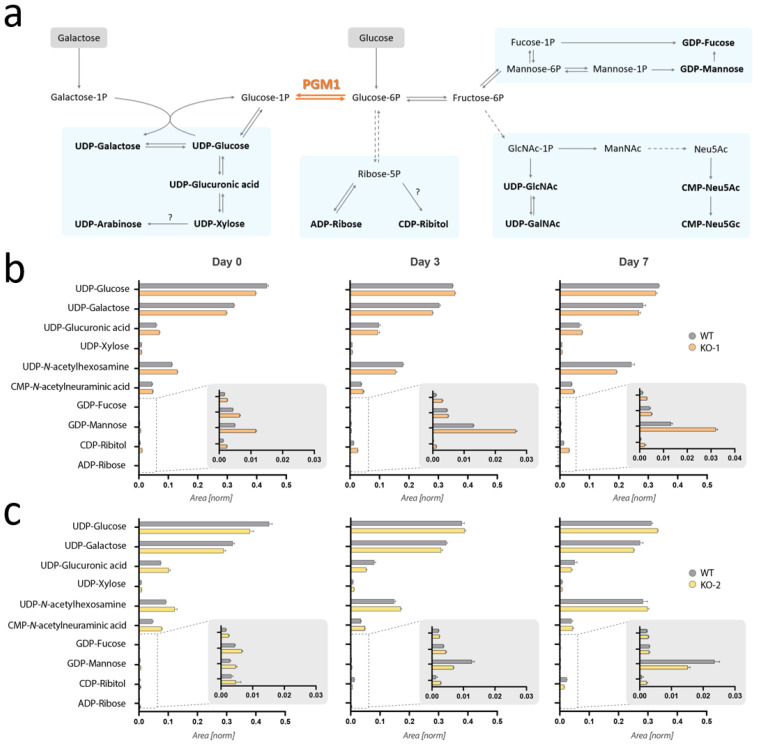
Steady-state metabolomics analysis of nucleotide sugars during myogenic differentiation. (**a**) Pathway scheme displaying the biosynthetic pathways of the main nucleotide sugars in human cells. Different pathways are clustered according to the proximity in terms of synthesis. Four clusters are indicated in the figure by the blue boxes. Solid lines indicate a unique reaction (bi- or uni-directional), while dashed arrows indicated a series of two or more reactions (omitted in the figure). Question marks flanking arrows indicate reaction(s) not yet described in mice. Glucose and galactose are highlighted in red and yellow, respectively. (**b**) Bar charts representing the composition of the nucleotide sugar pool in wild-type (WT) and *Pgm1*-KO clone 1 (KO-1) myoblasts (Day 0), intermediate differentiation state (Day 3), and myotubes (Day 7). The bars represent normalized peak area (n = 3) with indication of the standard error of the mean. Significance was tested via Mann–Whitney test, but no significant differences between the two cell lines were found (Appendix A). (**c**) Bar charts representing the composition of the nucleotide sugar pool in wild-type (WT) and *Pgm1*-KO clone 2 (KO-2) myoblasts (Day 0), intermediate differentiation state (Day 3), and myotubes (Day 7). Data are presented as described for panel (**b**) and no significant changes were identified (Appendix A).

**Figure 4 ijms-24-08247-f004:**
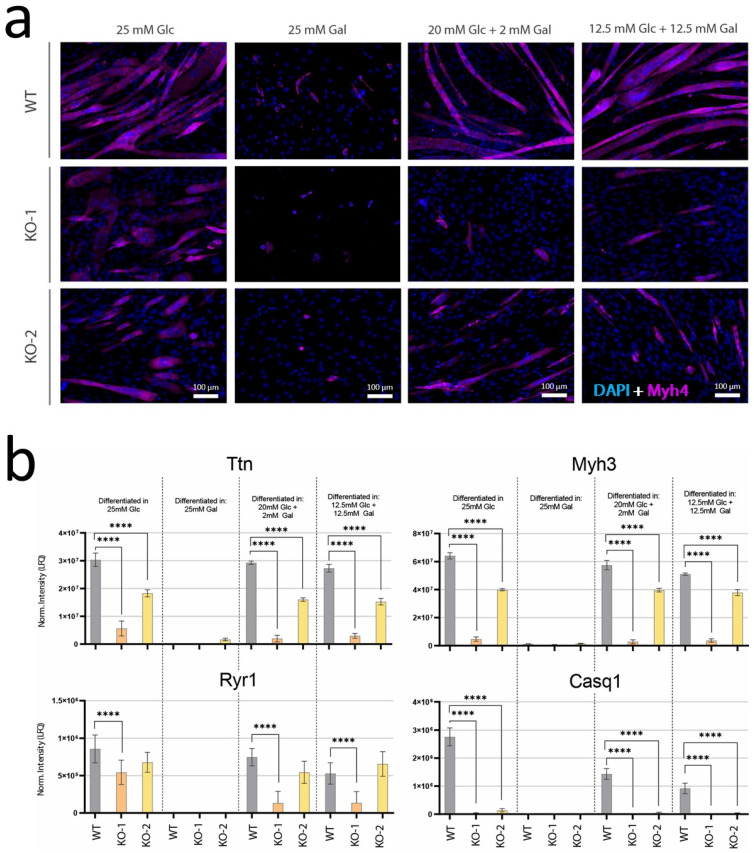
Myogenic maturation in presence of galactose. (**a**) Immunofluorescent staining of wild-type and *Pgm1*-KO myotubes (D7) differentiated in four differentiation media, containing: 25 mM glucose (control), 25 mM galactose, 20 mM glucose + 2 mM galactose, 12.5 mM glucose + 12.5 mM galactose. Blue, DAPI; magenta, Myosin (Myh4). Size-bar: 100 um. (**b**) Normalized protein levels of four myogenic maturation markers, Ttn, Myh3, Ryr1, and Casq1, in wild-type and KO myotubes (D7) differentiated with the same media reported in panel (**a**). Significance was tested via one-way ANOVA with Bonferroni post-hoc correction for multiple comparisons (only significant differences are shown: **** *p* < 0.0001; n = 10).

**Figure 5 ijms-24-08247-f005:**
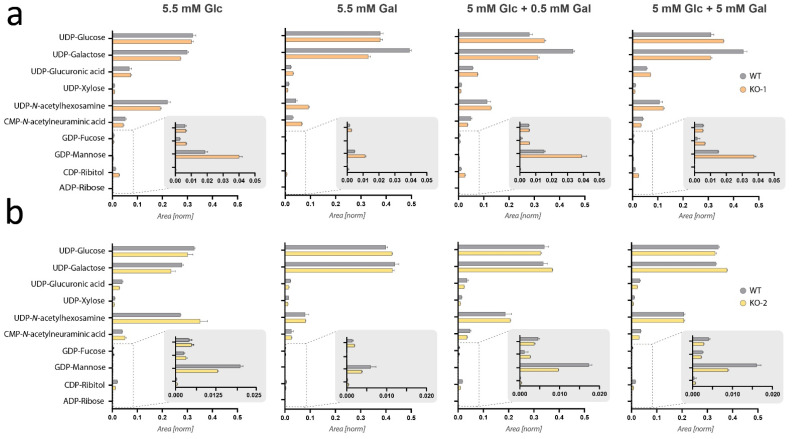
Steady-state metabolomics analysis of nucleotide sugars after galactose supplementation with or without glucose. (**a**) Bar charts displaying the composition of the nucleotide sugar pool in wild-type (WT) myotubes and *Pgm1*-KO clone 1 (KO-1) myotubes after 24 h incubation in different culture conditions that differ based on the concentration of galactose (Table 2). (**b**) Bar charts displaying the composition of the nucleotide sugar pool in wild-type (WT) myotubes and Pgm1-KO clone 2 (KO-2) myotubes after 24 h incubation in different culture conditions (Table 2). Bars represent normalized peak area (n = 3) and error bar based on the standard error of the mean. Significance was tested via Mann–Whitney test but no significant differences in panel (**a**) nor (**b**) between the two cell lines were found (Appendix A).

**Figure 6 ijms-24-08247-f006:**
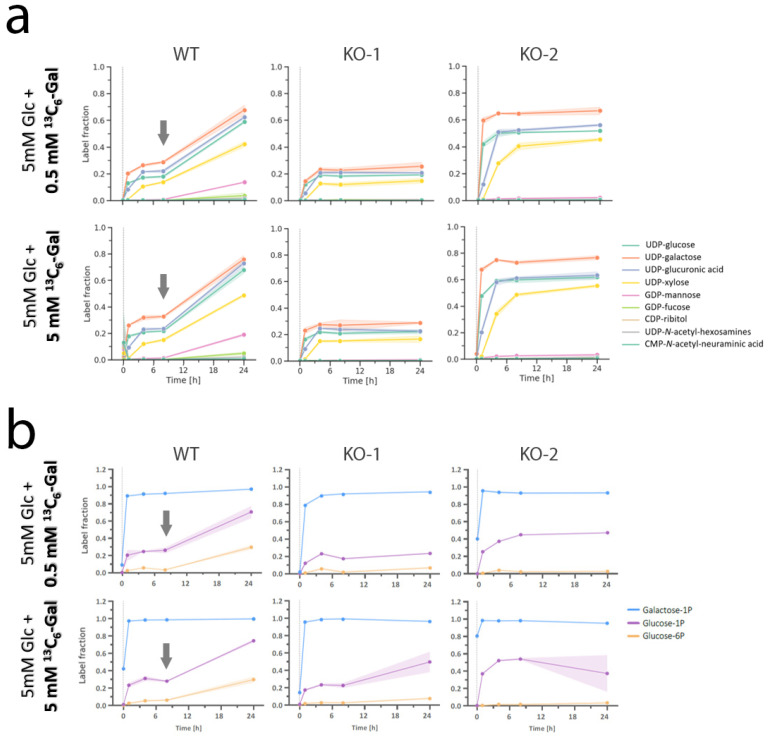
Metabolic flux analysis indicates a lower metabolic plasticity of *Pgm1*-KO clones. Metabolic flux into sugar moiety from tracing metabolomics analysis of selected nucleotide sugars (**a**) and phosphate hexoses (**b**) in wild-type and *Pgm1*-KO myotubes. The line charts represent the amount of labelled molecules in the pool of each metabolite. In the plots, a fraction of 1.0 indicates that the sugar moiety is labelled in all the molecules of the specific metabolite pool [20]. The concentration of ^13^C_6_-galactose in each culture condition is shown in the figure highlighted in red color. Data were normalized based on the ratio between the labelled fraction and the total fractions of the respective nucleotide sugar, ^13^C/^13^C + ^12^C. *y*-axis: labelled fraction. *x*-axis: time (hours).

**Figure 7 ijms-24-08247-f007:**
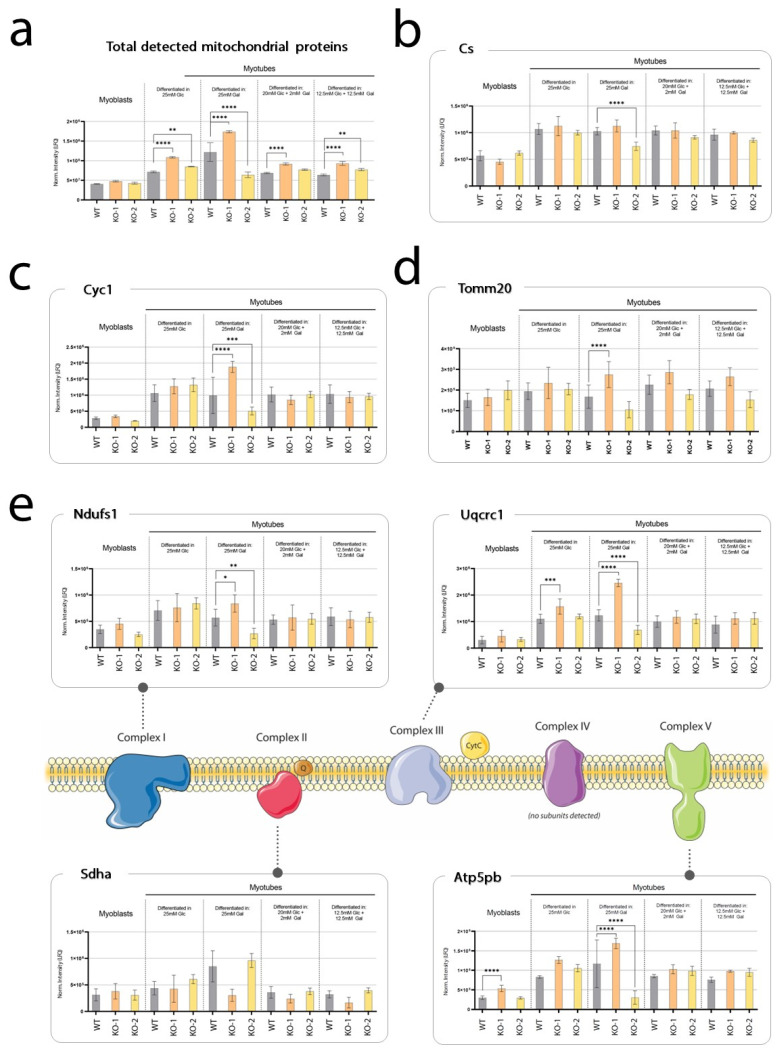
Levels of mitochondrial protein markers in wild-type and KO clones, at myoblast (D-1) and myotube state (D7). (**a**) Total sum of all mitochondrial proteins detected via MS-based proteomics. Significance was evaluated via one-way ANOVA with Bonferroni post-hoc correction for multiple comparisons (only significant differences are shown: * *p* < 0.05, ** *p* < 0.01, *** *p* < 0.001, **** *p* < 0.0001; n = 10). Normalized protein levels of citrate synthase (Cs) (**b**), cytochrome C (Cyc1) (**c**), Tomm20 (**d**), and of one subunit for each complex for the oxidative phosphorylation chain (**e**), with the exception of complex IV for which no subunits were efficiently detected. Significance was tested as for data reported in panel (**a**). Legend: grey, wild-type; orange, KO clone 1; yellow, KO clone 2.

**Figure 8 ijms-24-08247-f008:**
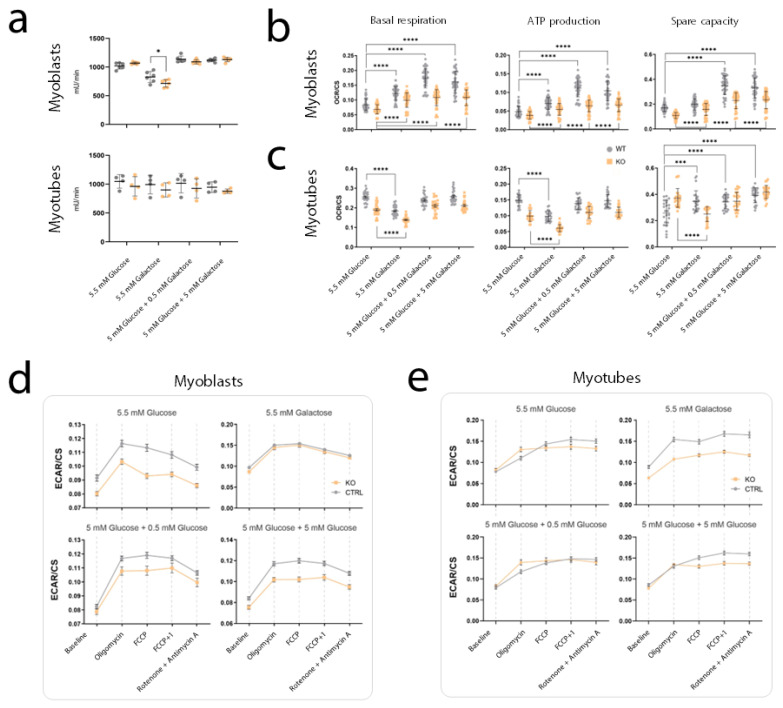
Bioenergetics parameters related to energy production resulting from MST of wild-type vs. *Pgm1*-KO myoblasts and myotubes in presence of galactose supplementation. (**a**) Citrate synthase activity measured in myoblasts and myotubes on both cell lines, these values were used to normalize the MST measurements recorded in myoblasts (**b**) and myotubes (**c**), both wild-type and KO-1. Statistical significance was calculated via one-way ANOVA with Bonferroni post-hoc correction for multiple comparisons (only significant differences are show in figure: * *p* < 0.05, *** *p* < 0.001, **** *p* < 0.0001). The horizontal line in the center of each cloud represents the mean, while the vertical line indicates the standard deviation of the mean. (**d**,**e**) ECAR values measured during MST in wild-type vs. KO-1 myoblasts and myotubes across all conditions.

**Table 1 ijms-24-08247-t001:** Results of the Pgm enzyme activity assay.

**Myoblasts**	**PGM** **mU/mg Protein**	**PMI** **mU/mg Protein**	**Ratio PGM/PMI**
**WT**	233.5	31.8	7.4
**KO clone 1**	12.2	36.6	0.3
**KO clone 2**	6.6	25.6	0.3
**Myotubes**	**PGM** **mU/mg protein**	**PMI** **mU/mg protein**	**Ratio PGM/PMI**
**WT**	353.8	77.2	4.6
**KO clone 1**	17.2	74.2	0.2
**KO clone 2**	8.4	47.9	0.2
**Dermal Fibroblasts** **(Reference)**	**PGM** **mU/mg protein**	**PMI** **mU/mg protein**	**Ratio PGM/PMI**
**Control 1**	225.1	47.9	4.7
**Control 2**	240.7	38.9	6.2
**Control 3**	310.0	39.5	7.8
**Patient 1**	10.4	36.9	0.2
**Patient 2**	8.9	47.3	0.3
**Patient 3**	11.5	38.9	0.3

**Table 2 ijms-24-08247-t002:** Composition of culture media and experimental media.

	Experiment	Type of Medium	Pyruvate	FBS Dialyzed	[Glc]	[Gal]	[Glc]:[Gal]
	Myoblast culture medium	DMEM [+Gln, 1%P/S]	1 mM	10%	25.0 mM	-	-
	Myogenic differentiation medium	DMEM [+Gln, 1%P/S]	1 mM	2%	25.0 mM	-	-
	Myotube culture medium	DMEM [+Gln, 1%P/S]	1 mM	2%	5.5 mM	-	-
Gene expression during differentiation	Treated differentiation medium 1	DMEM [+Gln, 1%P/S]	1 mM	2%	25.0 mM	-	1:0
	Treated differentiation medium 2	DMEM [+Gln, 1%P/S]	1 mM	2%	-	25.0 mM	0:1
	Treated differentiation medium 3	DMEM [+Gln, 1%P/S]	1 mM	2%	20.0 mM	2.0 mM	10:1
	Treated differentiation medium 4	DMEM [+Gln, 1%P/S]	1 mM	2%	12.5 mM	12.5 mM	1:1
Steady-state analysis Flux analysis	Myotube treatment medium 1	DMEM [+Gln, 1%P/S]	1 mM	2%	5.5 mM	-	1:0
	Myotube treatment medium 2	DMEM [+Gln, 1%P/S]	1 mM	2%	-	5.5 mM	0:1
	Myotube treatment medium 3	DMEM [+Gln, 1%P/S]	1 mM	2%	5.0 mM	0.5 mM	10:1
	Myotube treatment medium 4	DMEM [+Gln, 1%P/S]	1 mM	2%	5.0 mM	5.0 mM	1:1
MST in myoblasts	Myoblast treatment medium 1	DMEM [+Gln, 1%P/S]	-	10%	5.5 mM	-	1:0
	Myoblast treatment medium 2	DMEM [+Gln, 1%P/S]	-	10%	-	5.5 mM	0:1
	Myoblast treatment medium 3	DMEM [+Gln, 1%P/S]	-	10%	5.0 mM	0.5 mM	10:1
	Myoblast treatment medium 4	DMEM [+Gln, 1%P/S]	-	10%	5.0 mM	5.0 mM	1:1
MST in myotubes	Myotube treatment medium 1	DMEM [+Gln, 1%P/S]	-	2%	5.5 mM	-	1:0
	Myotube treatment medium 2	DMEM [+Gln, 1%P/S]	-	2%	-	5.5 mM	0:1
	Myotube treatment medium 3	DMEM [+Gln, 1%P/S]	-	2%	5.0 mM	0.5 mM	10:1
	Myotube treatment medium 4	DMEM [+Gln, 1%P/S]	-	2%	5.0 mM	5.0 mM	1:1

## Data Availability

Supplementary data are reported in the Appendix A. For raw data sharing, please contact the corresponding author.

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
