# Peer review of "In Vitro Skeletal Muscle Model of PGM1 Deficiency Reveals Altered Energy Homeostasis"

_ijms, 2023, doi:10.3390/ijms24098247_

Round 1

Reviewer 1 Report

Comments: 

Federica and colleagues established an in vitro mouse muscle cell model to study the muscle-specific metabolic mechanisms in PGM1 deficiency. They found galactose was unable to restore the reduced energy production capacity. Although the authors presented data relevantly support their conclusion, there are a couple of concerns that needs to be addressed.

1. Please correct errors and typos. For example, in line 155 of page 4, “clone 2 cells showed increased expression of Myog, Mef2 and Dmd over time”. Mef2 or Mef2c?

2, in figure 4, authors evaluate the relative expression of the muscle markers Myog, Mef2c and Dmd during differentiation (day 3) and at myotube state (day 7). They claim galactose supplementation in the culture medium, in combination with glucose, does not show clear effects on myogenic differentiation. However, myog and dmd increased in both ko1 and ko2 cell lines. Does it indicate a better differentiation in KO cell line? Should the authors give any explanation? 

3, in figure 6a, authors performed metabolic flux analysis. how to understand the fraction ratio only reach to 30% even after 1 day cultivation in ko 1, while the other two cell lines reach 70%? For the 13c6-gal tracing study, metabolites should have different label pattern, for example, M, M+1, M+2, M+3….M+6. Which isotopologues do authors choose in figure 6a? According to Figure 6a, there is a huge difference between ko-1 and ko-2 cell line. In seahorse study (Figure 6b), authors found Pgm1-ko have lower energy generating capacity based on the result of ko-1 cell line. Why didn’t the author show the result of ko-2 cell line?

4, For the metabolomic study, how do the author perform the normalization? Total protein? Cell number?

Author Response

Dear Revierwer 1 and Editor,

We would like thank you for your revision and for helping us improving our manuscript.

Please, find hereinbelow the rebuttal to the revision (original comment in italics, each followed by our response).

With Kind Regards,

Dr. Federica Conte

Prof. Dirk J. Lefeber

----------------------------------------------------------------------

  1. Please correct errors and typos. For example, in line 155 of page 4, “clone 2 cells showed increased expression of Myog, Mef2 and Dmd over time”. Mef2 or Mef2c?

The text was thoroughly revised, and the typos were corrected, including the correction of Mef2 to Mef2c.

2, in figure 4, authors evaluate the relative expression of the muscle markers Myog, Mef2c and Dmd during differentiation (day 3) and at myotube state (day 7). They claim galactose supplementation in the culture medium, in combination with glucose, does not show clear effects on myogenic differentiation. However, myog and dmd increased in both ko1 and ko2 cell lines. Does it indicate a better differentiation in KO cell line? Should the authors give any explanation?

We agree that there is a difference between the wildtype and knockout lines in differentiation. This section aimed to study the effect of galactose supplementation within a single cell line, either wild-type, KO-1 or KO-2. When comparing the relative expression of the Myog and Dmd genes, we did not observe an effect of galactose supplementation on the differentiation markers within the individual cell lines. This might have been confusing from the text, for which reason we have adapted this section of the results. Moreover, we performed mass spectrometry-based proteomics analysis to investigate the protein levels of more differentiation markers (see Figures 2 and 4, and related paragraphs).

3, in figure 6a, authors performed metabolic flux analysis. how to understand the fraction ratio only reach to 30% even after 1 day cultivation in ko 1, while the other two cell lines reach 70%? For the 13c6-gal tracing study, metabolites should have different label pattern, for example, M, M+1, M+2, M+3….M+6. Which isotopologues do authors choose in figure 6a? According to Figure 6a, there is a huge difference between ko-1 and ko-2 cell line. In seahorse study (Figure 6b), authors found Pgm1-ko have lower energy generating capacity based on the result of ko-1 cell line. Why didn’t the author show the result of ko-2 cell line?

Concerning the tracer metabolomics experiment (Figure 6a), we consider two isotopologues (M+6, M+11) to detect a direct flux from 13C6-labelled galactose into the hexose part of the respective UDP-sugars as displayed in Figure 6a. We agree that this was not sufficiently clear, and we have adapted the description in the Methods (paragraph 4.7.2) and Figure 6 legend accordingly.

We agree that the difference in plateau labeling between KO1 and KO2 is clear, however, the most striking and consistent difference between the two knockout clones and the WT cells was the observation that both knockouts don’t display the ability to fuel galactose into glucose 6-phosphate and glycolysis after 8 hours of incubation.  To clarify this point, we additionally measured the label integration in galactose 1-phosphate, glucose 1- and 6-phosphate (Figure 6b), which further support this lack of metabolic plasticity in the KO clones.

Concerning the Seahorse XF mitochondrial stress assays (Figure 8b), we chose the KO clone with the lowest differentiation capacity (see revised Figure 2 and Paragraph 2.2), as in our view this represents the clone with the most significant phenotype. This has been better clarified in Paragraph 2.7 (lines 353-355) of the revised file.

4. For the metabolomic study, how do the author perform the normalization? Total protein? Cell number?

As reported in the Methods section, Paragraph 2.5.2 “Resulting peak areas were normalized (i) to the sum of all nucleotide sugars detected to obtain relative abundances in steady-state experiments, or (ii) to the ratio of labelled vs. total fractions in the metabolic flux studies (13C/12C+13C)”. We use two different ways of normalization, depending on the mode of analysis (steady-state versus tracer metabolomics) as reported before in Scherpenzeel et al. 2022 [reference 24].

Reviewer 2 Report

Federica, C., et al have evaluated the function of Phosphoglucomutase 1 (PGM1) and regulation of D-galactose on nucleotide sugars and energy metabolism in genome-edited C2C12 myoblasts using the CRISPR/Cas9 system. The authors managed to demonstrate the first in vitro muscle-specific C2Cl2 model by knocking out mouse pgm1 (homolog of human PGM1). They showed no depletion of UDP-Glc and UDP-Gal in both WT and KO cell types. Importantly, no major effects of galactose supplementation on myogenic differentiation. However, they showed a defect in galactose utilization for energy production by using a dynamic tracing of 13C6-galactose in myotubes and showed deterioration of ATP production and basal respiration in both myoblasts and myotubes after galactose supplementation, which needs further validation.

The experiments designed for this study are justified but are not well explained. Overall, the results are significant but there is a lack of experimental data to justify the conclusions which need more experiments with appropriate controls to be performed to validate the utilization of galactose as a metabolic substrate for energy production and more importantly to show the aberrant energy homeostasis in PGM1-deficient skeletal muscle cell lines. Many other flaws include a lack of scientific data and its principles. Authors have missed explaining several facts which had been already mentioned in previously published journals related to mitochondrial activity and membrane potential as they mentioned mitochondrial damage and proton leak in their discussion. These effects could be easily shown by doing simple mitochondrial membrane potential assays in the myotube and myoblasts which could give us a better vision of cell apoptosis (Yokoyama, S. et al., 2020).

Nonetheless, the article seemed to possess some major concerns for myotube and/or myoblast's structural prediction. The authors used bright field light microscopy to show the growth of the cell lines and cell structure. On the better side, high-resolution microscopy would provide more information and additional structural information. The novelty of this study is the pathogenesis of PGM1 deficiency in vitro muscle-specific C2C12 model by knocking out mouse Pgm1 (homolog of human PGM1) and demand of galactose as a substrate for energy metabolism. The authors have not described the structures and the number of mitochondria involved in the comparison of the cell lines. which could be easily performed by simple IHC and western blots (to check the protein levels of different mitochondrial complexes and/or outer & inner membrane proteins) as the cell lines are easily culturable and available. This data will give us another set of valuable information on muscular energy homeostasis, which could be highly relevant to this study.

Overall, the clarity of the text is good but needs some explanation and readjustments. The manuscript has very few typographical and grammatical errors. A few main figures require a bit of attention on light adjustments and graphical numbering. The quantitative analyses are much appreciated. The authors need to describe some of the results, especially on the expression of genes involved in myotube differentiation, nucleotide sugars, and energy metabolism. In general, the manuscript can accomplish the caliber of quality for consideration for publication in the International Journal of Molecular Sciences with some major details. Although a careful rewrite is necessary with new data the conclusions may change. The authors are advised to consider the comments below:

Major comments

1.      Results / 2.1. Generation of Pgm1-knockout C2C12 myoblasts as muscle model for PGM1 deficiency / There is no concrete evidence provided against Pgm1-knockout C2C12 myoblasts. A simple western blot or immunofluorescence analysis of some of the markers like an expression of intercellular adhesion molecule 1 (ICAM-1), would show that the KO cell model mimics PGM-1 deficient patients. And provide some supporting data like Glucose quantification in the cells, Pgm-1 expression, and activity in the cells before moving forward.

2.      In Figure 2b / The morphology of Pgm1-KO clones and wild-type, cultured myoblasts need to show in high-resolution images. It is very difficult to view the morphological differences of the mature myotubes on day 7 of differentiation. A recommendation of proper nuclei staining like Giemsa staining (Barbieri, E. et al., 2011) should use to view multinucleation or even with fluorescence phalloidin staining for actin filament / DAPI will provide clear structural details between WT vs KO clones.

3.      In Figure 2c / Relative levels of UDP-N-acetyl hexosamines are highly increased (almost twice the level of expression on Day 7 comparing Day 0) over differentiation in WT compared to the KO cell lines. On the other hand, CMP-N-acetyl-neuraminic acid, was only increased in one of the KO lines (KO2) at Day 0, and why? Even the expression of CDP-Ribitol was very high in KO-1 and very low in KO-2. This part of the figure needs a detailed explanation as the results do seem not consistent. Supporting statistical data (excel sheet) would help to support the statements that the author claims.

4.      Results / 2.4. Galactose treatment does not affect myogenic differentiation / To compare the growth rate for WT and KO in different conditions, 1 more condition would be more crucial “no added carbohydrate condition” to explain that galactose and no carbohydrate medium have less growth rate as compared to Glu + Gal and high glucose condition means these cells do not Substitute Glucose with Galactose as energy substrate.

5.      In Figure 4a / The morphology of Pgm1-KO clones and wild-type, cultured myoblasts need to show in high-resolution images. Especially, to see the structural difference in the KO-2 cell line cultured with 25mM Galactose. Why does the Pgm1-KO clone 2 reach full confluence also fig 4b – why does Pgm1-KO clone 2 showed 4 times higher expression of Myog (one of the myogenic differentiation markers) compared to WT when cultured with 25mM Galactose?

6.      Results / 2.4. Galactose treatment does not affect myogenic differentiation / Also, a measurement of the consumption of galactose under different growth conditions in all the cell lines would provide us clarity on the needs of Galactose for energy metabolism.  As if Galactose is not being consumed the concentration should remain the same throughout the assay in the culture medium.

7.      Results / 2.5. Effect of galactose treatment on nucleotide sugar levels / Considering the galactose supplementation, adding 0.5mM Galactose gives the same effect comparing the addition of 5mM or 5.5mM Galactose in WT. But there is no significant reduction of UDP-Glu in the WT in all 4 conditions. On the other hand, Figure 5b / The graphs do not support the statement that “In all conditions in which glucose is present in culture, the level of UDP-Gal remain lower than in control”, which needs more explanation.

8.      Results / 2.6. Dynamic tracing of 13C-labelled galactose shows reduced galactose utilization in Pgm1-KO clones / First of all, to compare galactose utilization between WT vs KO clones in different conditions, 2 more conditions would be more crucial “0.5 mM Glc + 0.5 mM 13C6-Gal & only 0.5 mM 13C6-Gal” which will provide a more clear idea about the metabolic shift and utilization of galactose to fuel nucleotide sugar synthesis in wild-type myotubes compared to KO clones.

Secondly, based on the experiment it seems the KO-clone 2 (where PGM1 is deficient) also showed metabolic flexibility which is even better than WT as the levels of nucleotide sugar are very stable from 2 hours time point. It would be very helpful if you could explain these results where it seems that KO clone 2 is more efficient than WT in metabolizing galactose according to Figure 5a.

9.      Results / 2.7. Pgm1-KO myoblasts and myotubes have a lower energy-generating capacity, not affected by galactose treatment / The statement “The addition of Gal in combination with Glc increased the basal respiration of both wild-type and Pgm1-KO myoblasts and less dramatically in myotubes” is not true. According to the graphical representation, the level of basal respiration in Pgm1-KO myoblasts is almost the same in all conditions where Gal is present and is more in Pgm1-KO myotubes when Gal is combined with Glc in the medium. It would be helpful if you could explain these results carefully.

10.   Based on Seahorse Assay a lot of speculation could be made like if there is a higher proton leak that indicates dysfunctional mitochondria or low maximum respiratory capacity means unable to or slow in rapid oxidation of substrates etc. But what if there is a smaller number of mitochondria in KO cell lines? Or does the WT contain a double number of mitochondria with half their activity?

Simple mitochondrial staining and counting in WT vs Pgm1-KO myoblasts/myotubes will provide good evidence against mito number. Even with high-resolution imaging the structure of mitochondria will provide more perspective like does the Pgm1-KO myoblasts/myotubes contain fragmented and/or damaged mitochondria compared to the WT, which will explain the reason behind global lower basal respiration in the Pgm1-KO myoblasts/myotubes.

11.   It is important to measure the cell respiratory capacity in a glucose-free assay environment with only pyruvate and amino acids available as energy sources for WT and Pgm1-KO myoblasts/myotubes.

12.   I also suggest assessing respiratory chain activities. A muscle cell line would have high metabolic rates and accessibility and the assessment of mitochondrial respiratory chain (RC) enzymatic activities is essential for investigating mitochondrial function in several metabolic situations. Especially, mitochondrial complex I to V. (Spinazzi, M. et al., 2012, Frazier, A E. et al., 2020, Brischigliaro, M. et al., 2022, Turton, N. et al., 2022)

Minor comments

1.      In Figure 2c / “Clone 1 /2 cells showed increased expression of all markers over differentiation.” It will be helpful for a clear understanding if you provide all the markers used over differentiation with their expression curve or chart (even as if in supplementary). 

2.      In Figure 2c / All the graphs in each panel are inconsistent based on labeling (fold change).  Please do the labeling properly and a statistical analysis would give us a clear perspective of the data.

3.      Figure 2C / All graphs / check labeling D1, not D-1

4.      In Figure 4b / All the graphs in each panel are inconsistent based on labeling (fold change).  Please do the labeling properly and provide a statistical analysis (especially the expression of Dmd at D7) would give us a clear perspective of the data.

5.      Sup Fig 3 / text line 6 / check spelling “proton leak”

Author Response

Dear Reviewer 2 and Editor,

We would like thank you for your thorough revision that pushed us to improve the quality of our manuscript.

Please, find hereinbelow the rebuttal to the revision (original comment in italics, followed by our response).

With Kind Regards,

Dr. Federica Conte

Prof. Dirk J. Lefeber

---------------------------------------------------------------------------------

1. Results / 2.1. Generation of Pgm1-knockout C2C12 myoblasts as muscle model for PGM1 deficiency / There is no concrete evidence provided against Pgm1-knockout C2C12 myoblasts. A simple western blot or immunofluorescence analysis of some of the markers like an expression of intercellular adhesion molecule 1 (ICAM-1), would show that the KO cell model mimics PGM-1 deficient patients. And provide some supporting data like Glucose quantification in the cells, Pgm-1 expression, and activity in the cells before moving forward.

We agree with the reviewer that evidence of true knockouts is important. We aimed to show this by indicating Pgm1 enzyme deficiency in the C2C12 knockout clones (old Figure 2A, now Table 1). The residual levels are highly comparable to the levels in fibroblasts of confirmed PGM1 deficient patients, which commonly show an absence of PGM1 protein by western blot (reference [2] in the paper). To further stress the similarity with PGM1-deficient patients, we included enzyme activity measurements from patient derived primary dermal fibroblasts in Table 1 of the revised version.

Additionally, we performed an additional experiment using MS-based proteomics to confirm the effect of CRISPR/Cas9 genome editing on PGM1 protein levels. Expression proteomics detected the presence of Pgm1 protein was readily identified in WT C2C12 myoblasts and myotubes, while the protein was completely absent in both knockout clones. We have added this information to the Results section and in Figure 2b in the revised version.

The addition of the results from the proteomics analysis, along with the results from the Sanger sequencing (Supp. Figure 1) and the enzyme activity assay (Table 1), confirm the successful editing of mouse Pgm1 gene and consequential complete loss of function.

2. In Figure 2b / The morphology of Pgm1-KO clones and wild-type, cultured myoblasts need to show in high-resolution images. It is very difficult to view the morphological differences of the mature myotubes on day 7 of differentiation. A recommendation of proper nuclei staining like Giemsa staining (Barbieri, E. et al., 2011) should use to view multinucleation or even with fluorescence phalloidin staining for actin filament / DAPI will provide clear structural details between WT vs KO clones.

We agree with the reviewer to provide additional data to visualize the morphological differences. We have performed novel differentiation experiments and provide images with higher resolution and magnifications both in the main figures (Figures 2 and 4) and in the new Supp. Figure 3. Additionally, we performed additional immunofluorescence staining using DAPI as nuclear staining and a fluorescent antibody against Myosin Heavy Chain 4 (Myh4) to provide more insights into the structure and maturation state of the WT and KO clones during differentiation (Figures 2 and 4). In the revised file, we have adapted the Result section, Methods section, Figures and captions accordingly. We believe that these novel data better show the effects of Pgm1-knockout on the morphology and myotube formation.

3. In Figure 2c / Relative levels of UDP-N-acetyl hexosamines are highly increased (almost twice the level of expression on Day 7 comparing Day 0) over differentiation in WT compared to the KO cell lines. On the other hand, CMP-N-acetyl-neuraminic acid, was only increased in one of the KO lines (KO2) at Day 0, and why? Even the expression of CDP-Ribitol was very high in KO-1 and very low in KO-2. This part of the figure needs a detailed explanation as the results do seem not consistent. Supporting statistical data (excel sheet) would help to support the statements that the author claims.

We agree that the levels of UDP-N-acetyl hexosamines (Figure 3c) increase during differentiation from myoblast at Day 0 to myotubes at Day 7. This is a general tendency that we observe during myoblast differentiations in our laboratory, and might be linked to the type of normalization we performed (normalization over the total peak area, as reported in van Scherpenzeel et al. 2022, reference [4]). In our view, we don’t see significant differences between the three cell lines (WT, KO1 and KO2) in our data, as confirmed by the Mann-Whitney test we performed to evaluate statistical significance (Supp. File 2). We have added the normalized steady-state metabolomics data and statistical analysis as Supp. File 2.

4. Results / 2.4. Galactose treatment does not affect myogenic differentiation / To compare the growth rate for WT and KO in different conditions, 1 more condition would be more crucial “no added carbohydrate condition” to explain that galactose and no carbohydrate medium have less growth rate as compared to Glu + Gal and high glucose condition means these cells do not substitute Glucose with Galactose as energy substrate.

We did not include a condition without carbohydrates as in our experience a complete lack of carbohydrates in the culture medium prevents the cells from growing/proliferating, and causes cell death. Besides, we aimed to study the effects of galactose on muscle cells in conditions that are (as much as possible) comparable with the physiological conditions in which patients’ skeletal muscle cells would be (galactose treatment in addition to a normal diet including glucose). A complete depletion of carbohydrates would represent a non-physiological situation in which galactose effects would not have a significant translational value.

5. In Figure 4a / The morphology of Pgm1-KO clones and wild-type, cultured myoblasts need to show in high-resolution images. Especially, to see the structural difference in the KO-2 cell line cultured with 25mM Galactose. Why does the Pgm1-KO clone 2 reach full confluence also fig 4b – why does Pgm1-KO clone 2 showed 4 times higher expression of Myog (one of the myogenic differentiation markers) compared to WT when cultured with 25mM Galactose?

To address this point, we included improved bright-field images and immunofluorescence staining of myotubes differentiated in the four combination of glucose and/or galactose in Figures 2 and 4, and in Supp. Figure 4.

Concerning the first question, the image we included in the previous version resulted misleading, as KO-2 cells do not usually reach confluence in this condition. Besides, also the other two lines, WT and KO-1, shown some cell survival when culture with 25 mM Gal. As we repeated the differentiation to perform additional experiments, we decided to collect new and more representative images to replace the one in the previous version (Figure 4, Supp. Figure 4). Nonetheless, we agree with the Reviewer that KO-2 cells seem to better survive in this condition compared to the other cell lines (Supp. Figure 4), which instead display more apoptotic-like morphology (e.g. flat cells with some extent of cellular fragmentation and formation of apoptotic bodies), although maturation fails in all the lines. We hypothesize that this different might be due to clonal differences.

Concerning the expression of MyoG, while reviewing the data during the revision process, we found inconsistencies in the results obtained from the RT-qPCR of this specific gene, and thus we removed these data from the manuscript, while we integrated it with MS-based proteomics that enabled us to investigate more effectively a wider number of differentiation markers (Figure 4). In this regard, the protein levels of four main markers of skeletal muscle maturation (Ttn, Ryr1, Myh3, Casq1) did not show any significant increase in KO-2 cells in condition 2 (25 mM Gal).

In conclusion, although this line seems to survive galactose feeding, seems to not show clear myotube maturation, as consistently shown by the levels of protein markers (Figure 4b), immunofluorescence staining (Figure 4a) and brightfield optic microscopy (Supp. Figure 3).

6. Results / 2.4. Galactose treatment does not affect myogenic differentiation / Also, a measurement of the consumption of galactose under different growth conditions in all the cell lines would provide us clarity on the needs of Galactose for energy metabolism. As if Galactose is not being consumed the concentration should remain the same throughout the assay in the culture medium.

The message we wished to get across with our Seahorse data is that Pgm1 deficiency results in lower energy production (ATP) which wasn’t restored by addition of galactose, since this is a main outstanding question for patients’ phenotypes upon dietary galactose supplementation. We agree that our data don’t provide direct evidence for the consumption of galactose in C2C12 cells. However, our data (specifically tracer metabolomic/Figure 6, and Seahorse MST/Figure 8) support the notion that galactose is unable to rescue the reduced ATP. Obviously, galactose can be/is used by the cell for other metabolic pathways as well, and therefore the levels of galactose in the culture medium will not remain constant. For this reason, we believe that the analysis of galactose consumption in the medium will provide additional insights in our specific question to the effect of galactose on ATP generation.

7. Results / 2.5. Effect of galactose treatment on nucleotide sugar levels / Considering the galactose supplementation, adding 0.5mM Galactose gives the same effect comparing the addition of 5mM or 5.5mM Galactose in WT. But there is no significant reduction of UDP-Glu in the WT in all 4 conditions. On the other hand, Figure 5b / The graphs do not support the statement that “In all conditions in which glucose is present in culture, the level of UDP-Gal remain lower than in control”, which needs more explanation.

We agree with the reviewer that there is no evidence for an absolute decrease in UDP-Glucose levels. However, we intended to refer to the relative levels of UDP-Glucose and UDP-Galactose. Comparison of relative levels in the WT cell line suggests that the ratio of UDP-Galactose/UDP-Glucose slightly increases upon feeding cells with galactose. This effect is not really seen in KO1, while it is slightly visible in KO2. Additionally, comparing the relative levels of UDP-Galactose and UDP-Glucose with UDP-HexNAcs levels also shows an effect of galactose feeding. In this respect, it is important to look at the relative levels. To make this message more clear, we have adapted the text of Results section, Paragraph 2.5.

8a. Results / 2.6. Dynamic tracing of 13C-labelled galactose shows reduced galactose utilization in Pgm1-KO clones / First of all, to compare galactose utilization between WT vs KO clones in different conditions, 2 more conditions would be more crucial “0.5 mM Glc + 0.5 mM 13C6-Gal & only 0.5 mM 13C6-Gal” which will provide a more clear idea about the metabolic shift and utilization of galactose to fuel nucleotide sugar synthesis in wild-type myotubes compared to KO clones.

In our experience C2C12 cells do not grow/proliferate in such low carbohydrate concentrations, thus we won’t have the possibility to maintain viable cells in such conditions for 24 hours (time window necessary for our tracer metabolomics experiment). For this reason, we feel that the results of such metabolic flux experiments will be affected by poor cell growth and will not allow to draw reliable conclusions on the metabolic adaptation. Besides, we are interested in replicating in vitro physiological situations (comparable with the condition in which patients’ skeletal muscle cells need to function), which is the reason why we opted for maintaining 5 mM Glc concentration as base condition, on top of which galactose is added to simulate galactose supplementation.

8b. Secondly, based on the experiment it seems the KO-clone 2 (where PGM1 is deficient) also showed metabolic flexibility which is even better than WT as the levels of nucleotide sugar are very stable from 2 hours time point. It would be very helpful if you could explain these results where it seems that KO clone 2 is more efficient than WT in metabolizing galactose according to Figure 5a.

We believe that a misunderstanding on the definition of ‘metabolic flexibility’ has occurred. By ‘metabolic flexibility (or plasticity)’ we intended the ability of a cell line to switch from glucose to galactose metabolization when glucose levels in the medium progressively decreased due to consumption.

In the case of WT myotubes, before 8 hours only a small amount of the UDP-Glc, UDP-Gal, UDP-GlcA and UDP-Xyl are labelled, meaning only a small amount of the nucleotide sugars derives from 13C6-galactose, while the rest derives mostly from unlabeled glucose. After 8 hours, when the glucose in the medium is decreasing, the WT myotubes are able to adapt by progressively switching to galactose consumption to fuel the synthesis of these four nucleotide sugars. This is shown by the increase in the labelled fraction of these four nucleotide sugars’ pools, indicating that more of these molecules derived from 13C6-galactose. On the contrary, both KO clones (although with different efficiency) seems to rely only on 13C6-galactose for the synthesis of these nucleotide sugars since timepoint 0, likely because this metabolic switch cannot occur if Pgm1, which is the enzyme located at the cross-roads between glycolysis and nucleotide sugar synthesis, is deficient.

We clarified this point in the Result section of the revised manuscript, Paragraph 2.6.

9. Results / 2.7. Pgm1-KO myoblasts and myotubes have a lower energy-generating capacity, not affected by galactose treatment / The statement “The addition of Gal in combination with Glc increased the basal respiration of both wild-type and Pgm1-KO myoblasts and less dramatically in myotubes” is not true. According to the graphical representation, the level of basal respiration in Pgm1-KO myoblasts is almost the same in all conditions where Gal is present and is more in Pgm1-KO myotubes when Gal is combined with Glc in the medium. It would be helpful if you could explain these results carefully.

We adjusted the statistics showing significance between the control condition (5.5 mM Glc) and the other conditions, which we hope will make comparisons easier. Regarding the sentence, indeed it was not properly phrased, and in the revised version of the manuscript it has been corrected (lines 362-364).

10. Based on Seahorse Assay a lot of speculation could be made like if there is a higher proton leak that indicates dysfunctional mitochondria or low maximum respiratory capacity means unable to or slow in rapid oxidation of substrates etc. But what if there is a smaller number of mitochondria in KO cell lines? Or does the WT contain a double number of mitochondria with half their activity?

Simple mitochondrial staining and counting in WT vs Pgm1-KO myoblasts/myotubes  will provide good evidence against mito number. Even with high-resolution imaging the structure of mitochondria will provide more perspective like does the Pgm1-KO myoblasts/myotubes contain fragmented and/or damaged mitochondria compared to the WT, which will explain the reason behind global lower basal respiration in the Pgm1-KO myoblasts/myotubes.

We appreciate the concerns on mitochondria in relation to our findings. We had performed activity measurements of citrate synthase (Cs) to normalize our Seahorse data in the previous version (now more clearly explained in the Methods section) to correct for mitochondrial abnormalities. Furthermore, we performed extensive proteomics expression profiling and extracted data on mitochondrial proteins (Figure 7) from both myoblasts and myotubes differentiated in presence of different combination of glucose and galactose (see Table 2). Although the sum of all detected mitochondrial proteins seems to indicate a higher mitochondrial content in the KO lines, especially KO-1 (Figure 7a), markers of mitochondrial functionality and oxidative capacity, such as Cs and Cyc1, did not show significant differences among the three cells lines across the different conditions (with the exception of condition 2, in which the differentiation medium was supplemented with 25 mM galactose, in which cells were poorly maturing). For this reason, we decided to normalize the MST results on the Cs activity (results now reported in Figure 8a). These additional details have been included in the Results section, Paragraph 2.7, of the revised version.

11. It is important to measure the cell respiratory capacity in a glucose-free assay environment with only pyruvate and amino acids available as energy sources for WT and Pgm1-KO myoblasts/myotubes.

Our study aimed to replicate physiological conditions similar to those in which the skeletal muscle cells of patients would be exposed to in the body. Thus, testing conditions in which only pyruvate or amino acids are available for energy production would not reflect what could happen in vivo, besides being very difficult for these cells to survive without carbohydrates in culture (especially without glucose, see previous comments).

12. I also suggest assessing respiratory chain activities. A muscle cell line would have high metabolic rates and accessibility and the assessment of mitochondrial respiratory chain (RC) enzymatic activities is essential for investigating mitochondrial function in several metabolic situations. Especially, mitochondrial complex I to V. (Spinazzi, M. et al., 2012, Frazier, A E. et al., 2020, Brischigliaro, M. et al., 2022, Turton, N. et al., 2022).

In line with the previous questions, we agree that mitochondrial dysfunction could affect Seahorse results. We extracted protein information on several subunits of respiratory chain complexes and found mostly comparable levels between WT and KO lines (with the exception of condition 2, in which cells are fed with only galactose, that showed significant variability between cells lines, likely due to compromised maturation) (Figure 7). Of course, our present study does not aim to fully explain the mitochondrial mechanisms that take place as consequence of Pgm1 loss of function, but rather to set the first foundation for future studies dedicated to the investigation of the mitochondrial state and oxidative capacity of Pgm-deficient muscle models.

Minor comments:

1. In Figure 2c / “Clone 1 /2 cells showed increased expression of all markers over differentiation.” It will be helpful for a clear understanding if you provide all the markers used over differentiation with their expression curve or chart (even as if in supplementary). 

After proteomics analysis, we included the protein levels of additional differentiation markers (Figure 2d), and plot four main skeletal muscle markers (Figure 2c), namely titin (Ttn), ryanodine receptor (Ryr1), calsequestrin (Casq1) and myosine heavy chain 3 (Myh3). Moreover, all the data have been included in Supp. File 1.

2.In Figure 2c / All the graphs in each panel are inconsistent based on labeling (fold change).  Please do the labeling properly and a statistical analysis would give us a clear perspective of the data.

The statistical significance was evaluated via one-way ANOVA with Bonferroni correction (see revised Figure 2c). Regarding the inconsistent label, we believe the Reviewer meant the different scales on the X-axis. Due to the wide difference between the expression fold-change of the markers displayed, we chose to visualize the markers separately to avoid compromising the visibility of some of the bars. Log-scale representation was tried but did not fully aid the visualization. However, we hope that the addition of the statistical significance, when present, can help clarifying the visualization.

3. Figure 2C / All graphs / check labeling D1, not D-1

The labelling D-1 is correct. D-1 (day -1) refers to one day prior to starting the experiment, The main difference is that at day -1 (24h before starting the experiment) the myoblasts are around 70-75% confluent, and did not start spontaneous differentiation yet. Vice versa, at day 0 (D0), when we start the experiment, the myoblasts are 100% confluent and already start showing the first signs of spontaneous differentiation (e.g. cell elongation, sporadic fusion, cell alignment). Thus, at full confluence these cells cannot be considered as ‘full’ myoblasts anymore. For this reason,  we collected myoblasts one day prior to full confluence.

4. In Figure 4b / All the graphs in each panel are inconsistent based on labeling (fold change). Please do the labeling properly and provide a statistical analysis (especially the expression of Dmd at D7) would give us a clear perspective of the data.

In this case, we were able to display the markers in the same plot, solving the issue with the Y axis scale. Moreover, gene expression markers have been moved from Figure 4 to Supp. Figure 5, and replace in the main text by the protein levels of differentiation markers obtained via extensive MS-based proteomics analysis, as this analysis provided a larger number of markers and more informative data (see revised Figures 2 and 4).

5. Sup Fig 3 / text line 6 / check spelling “proton leak”

The typo has been addressed in the revised version.

Reviewer 3 Report

The manuscript "In vitro skeletal muscle model of PGM1 deficiency reveals altered energy homeostasis" by Conte et al is devoted to the establishing an in vitro muscle model for PGM1 deficiency to investigate the role of PGM1 and the effect of D-galactose on nucleotide sugars and energy metabolism. The manuscript as a whole and all its section are well written and structured. The Discussion is well organized and detailed. The manuscript may be accepted after minor revision 

1) The first paragraph in the introduction should be deleted. Probably the authors inserted it by mistake.

2) Figure 2A should be presented as a table, not a figure.

3) Figures 3-5 is not labeled to show the statistical significance of the differences.

4) Microsoft Office Excel is a not software for statistical analysis

5) Description of the statistics at the end of section is not optimal. It is worth making subsection “4.7. statistical analysis” in the end of the material and method section.

Author Response

Dear Reviewer 3,

We would like thank you for your revision and suggestions that helped us improving our manuscript.

Please, find hereinbelow the rebuttal to the revision (original comment in italics, followed by our response).

With Kind Regards,

Dr. Federica Conte

Prof. Dirk J. Lefeber

---------------------------------------------------------------------------------

1) The first paragraph in the introduction should be deleted. Probably the authors inserted it by mistake.

Indeed, that paragraph was part of the template file and not part of the article. This paragraph has been removed.

2) Figure 2A should be presented as a table, not a figure.

We agreed with the comment and adjusted the figure as requested (now Table 1 in the revised file).

3) Figures 3-5 is not labeled to show the statistical significance of the differences.

After testing the normality of the data distribution, we evaluated the statistical significance of the steady-state metabolomics data using Mann-Whitney test, which however did not detect any significance change. This information has been added to the captions of both Figure 3 and 5 in the revised file, and we have included the normalized steady-state metabolomics data and statistical analysis in Supp. File 2.

4) Microsoft Office Excel is a not software for statistical analysis

Indeed, data processing and statistical analysis was performed in GraphPAD Prism, while Excel was used to normalize the data exported from Skyline prior to import in GraphPAD. We adjusted the text accordingly.

5) Description of the statistics at the end of section is not optimal. It is worth making subsection “4.7. statistical analysis” in the end of the material and method section.

We understand the reason for the suggestion, however we believe it would be confusing to create such paragraph since different types of statistical tests were performed based on the experimental design, type of comparison, normality of the distribution, etc. However, we better clarify the type of statistical analysis performed for each experiment both in the Methods section and in the caption of each main Figure.

Round 2

Reviewer 1 Report

Authors addressed all my concerns.

Reviewer 2 Report

Federica, C., et al have discussed and re-evaluated some of the points which had been mentioned in the previous review of their manuscript. The addition of a table, and high-resolution images, in the manuscript is much appreciated.  It has been also appreciated that the authors are honest about their RT-PCR data. I think the authors provide enough revised data to fulfill the need of this manuscript like additional experiments using MS-based proteomics. They have coherently explained the major concerns mentioned in the previously revised version. Their writing is severely improved with proper explanation, clarification, and references. I would also appreciate the addition of extra data in supplementary support of this manuscript.

After careful revision, the manuscript now accomplished the caliber of quality for consideration for publication in the International Journal of Molecular Sciences.